# One After Another: Learning Incremental Skills for a Changing World

**Nur Muhammad (Mahi) Shafiullah**
mahi@cs.nyu.edu
New York University

**Lerrel Pinto**
lerrel@cs.nyu.edu
New York University

## Abstract

Reward-free, unsupervised discovery of skills is an attractive alternative to the bottleneck of hand-designing rewards in environments where task supervision is scarce or expensive. However, current skill pre-training methods, like many RL techniques, make a fundamental assumption – stationary environments during training. Traditional methods learn all their skills simultaneously, which makes it difficult for them to both quickly adapt to changes in the environment, and to not forget earlier skills after such adaptation. On the other hand, in an evolving or expanding environment, skill learning must be able to adapt fast to new environment situations while not forgetting previously learned skills. These two conditions make it difficult for classic skill discovery to do well in an evolving environment. In this work, we propose a new framework for skill discovery, where skills are learned one after another in an incremental fashion. This framework allows newly learned skills to adapt to new environment or agent dynamics, while the fixed old skills ensure the agent doesn't forget a learned skill. We demonstrate experimentally that in both evolving and static environments, incremental skills significantly outperform current state-of-the-art skill discovery methods on both skill quality and the ability to solve downstream tasks. Videos for learned skills and code are made public on: https://notmahi.github.io/disk.

## 1 Introduction

Modern successes of Reinforcement Learning (RL) primarily rely on task-specific rewards to learn motor behavior (Levine et al., 2016; Schulman et al., 2017; Haarnoja et al., 2018; Andrychowicz et al., 2020). This learning requires well behaved reward signals in solving control problems. The challenge of reward design is further coupled with the inflexibility in the learned policies, i.e. policies trained on one task do not generalize to a different task. Not only that, but the agents also generalize poorly to any changes in the environment (Raileanu et al., 2020; Zhou et al., 2019). This is in stark contrast to how biological agents including humans learn (Smith & Gasser, 2005). We continuously adapt, explore, learn without explicit rewards, and are importantly incremental in our learning (Brandon, 2014; Corbetta & Thelen, 1996).

Creating RL algorithms that can similarly adapt and generalize to new downstream tasks has hence become an active area of research in the RL community (Bellemare et al., 2016; Eysenbach et al., 2018; Srinivas et al., 2020). One viable solution is unsupervised skill discovery (Eysenbach et al., 2018; Sharma et al., 2019b; Campos et al., 2020; Sharma et al., 2020). Here, an agent gets access to a static environment without any explicit information on the set of downstream tasks or reward functions. During this unsupervised phase, the agent, through various information theoretic objectives (Gregor et al., 2016), is asked to learn a set of policies that are repeatable while being different from each other. These policies are often referred to as behavioral primitives or skills (Kober & Peters, 2009; Peters et al., 2013; Schaal et al., 2005). Once learned, these skills can then be used to solve downstream tasks in the same static environment by learning a high-level controller that chooses from the set of these skills in a hierarchical control fashion (Stolle & Precup, 2002; Sutton et al., 1999; Kulkarni et al., 2016) or by using Behavior Transfer to aid in agent exploration (Campos et al., 2021). However, the quality of the skills learned and the subsequent ability to solve downstream tasks are dependent on the unsupervised objective.

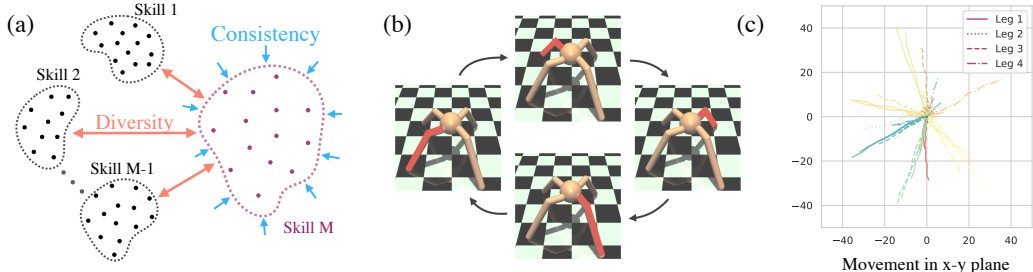

Figure 1: We present Discovery of Incremental Skills (DISk). As illustrated in (a), DISk learns skills incrementally by having each subsequent skills be diverse from the previous skills while being consistent with itself. DISk lets us learn skills in environment where dynamics change during training, an example of which is shown in (b): an Ant environment where a different leg breaks every few episodes. In this environment, DISk's learned skills' trajectories are shown in (c) under each broken leg – each color shows a different skill.

Unfortunately, current objectives for unsupervised skill discovery make a major assumption about the environment stationarity. Since all skills are simultaneously trained from start to finish, if the environment changes, every skill is equally exposed to highly non-stationary experience during its training. Moreover, all skills trying to adapt to the changed environment means that the agent may catastrophically forget previously learned skills, which is undesirable for an agent with expanding scope. This severely limits the application of current skill discovery methods in lifelong RL settings (Nareyek, 2003; Koulouriotis & Xanthopoulos, 2008; Da Silva et al., 2006; Cheung et al., 2020), a simple example of which is illustrated in Figure 1(b), or real-world dynamic settings (Gupta et al., 2018; Julian et al., 2020).

In this paper, we propose a new unsupervised model-free framework: Discovery of Incremental Skills (DISk). Instead of simultaneously learning all desired skills, DISk learns skills one after another to avoid the joint skill optimization problem. In contrast to prior work, we treat each individual skill as an independent neural network policy without parameter sharing, which further decouples the learned skills. Given a set of previously learned skills, new skills are optimized to have high state entropy with respect to previously learned skills, which promotes skill diversity. Simultaneously, each new skill is also required to have low state entropy within itself, which encourages the skill to be consistent. Together, these objectives ensure that every new skill increases the diversity of the skill pool while being controllable (Fig. 1(a)). Empirically, this property enables DISk to generate higher quality skills and subsequently improve downstream task learning in both dynamic and static environments compared to prior state-of-the-art skill discovery methods. Especially in evolving environments, DISk can quickly adapt new skills to the changes in the environment while keeping old skills decoupled and fixed to prevent catastrophic forgetting. We demonstrate this property empirically in MuJoCo navigation environments undergoing varying amounts of change. While DISk was designed for dynamic, evolving environments, we observe that it performs noticeably better than previous skill learning methods in static environments like standard MuJoCo navigation environments by improving skill quality and downstream learning metrics.

To summarize, this paper provides the following contributions: (a) we propose a new unsupervised skill discovery algorithm (DISk) that can effectively learn skills in a decoupled manner. (b) We demonstrate that the incremental nature of DISk lets agents discover diverse skills in evolving or expanding environments. (c) Even in static environments, we demonstrate that DISk can learn a diverse set of controllable skills on continuous control tasks that outperform current state-of-the-art methods on skill quality, as well as in sample and computational efficiency. (d) Finally, we demonstrate that our learned skills, both from dynamic and static environments, can be used without any modifications in a hierarchical setting to efficiently solve downstream long-horizon tasks.

## 2 BACKGROUND AND PRELIMINARIES

To setup our incremental RL framework for unsupervised skill discovery, we begin by concisely introducing relevant background and preliminaries. A detailed treatment on RL can be found in Sutton & Barto (2018), incremental learning in Chen & Liu (2018), and skill discovery in Eysenbach et al. (2018).

**Reinforcement Learning:** Given an MDP with actions $a \in A$, states $s \in S$, standard model-free RL aims to maximize a task specific reward $r_T(s, a)$ by optimizing the parameters $\theta$ of a policy $\pi_\theta$. In contrast to task-specific RL, where the rewards $r_T$ correspond to making progress on that specific task, methods in unsupervised RL (Bellemare et al., 2016; Eysenbach et al., 2018; Srinivas et al., 2020) instead optimize an auxiliary reward objective $r_I$ that does not necessarily correspond to the task-specific reward $r_T$.

**Incremental Learning:** Incremental, lifelong, or continual learning covers a wide variety of problems, both within and outside of the field of Reinforcement Learning. Some works consider the problem to fast initialization in a new task $T_{n+1}$ given previous tasks $T_i, 1 \le i \le n$ (Tanaka & Yamamura, 1997; Fernando et al., 2017), while others focus on learning from a sequence of different domains $D_1, \cdots, D_t$ such that at time $t + 1$, the model does not forget what it has seen so far (Fei et al., 2016; Li & Hoiem, 2017). Finally, other works have considered a bidirectional flow of information, where performance in both old and new tasks improves simultaneously with each new task (Ruvolo & Eaton, 2013; Ammar et al., 2015). In all of these cases, there is a distribution shift in the data over time, which differentiates this problem setting from multi-task learning. Some of these methods (Tanaka & Yamamura, 1997; Ruvolo & Eaton, 2013; Fei et al., 2016; Ruvolo & Eaton, 2013) learn a set of independently parametrized models to address this distribution shift over time.

**Skill Discovery:** Often, skill discovery is posited as learning a $z$-dependent policy $\pi_z(a|s; \theta)$ (Eysenbach et al., 2018; Sharma et al., 2020), where $z \in Z$ is a latent variable that represents an individual skill. The space $Z$ hence represents the pool of skills available to an agent. To learn parameters of the skill policy, several prior works (Hausman et al., 2018; Eysenbach et al., 2018; Campos et al., 2020; Sharma et al., 2020) propose objectives that diversify the outcomes of different skills while making the same skill produce consistent behavior.

One such algorithm, DIAYN (Eysenbach et al., 2018), does this by maximizing the following information theoretic objective:

$$\mathcal{F}(\theta) = I(S; Z) + \mathcal{H}(A|S, Z) \tag{1}$$

Here, $I(S; Z)$ represents the mutual information between states and skills, which intuitively encourages the skill to control the states being visited. $\mathcal{H}(A|S, Z)$ represents the Shannon entropy of actions conditioned on state and skill, which encourages maximum entropy (Haarnoja et al., 2017) skills. Operationally, this objective is maximized by model-free RL optimization of an intrinsic reward function that corresponds to maximizing Equation 1, i.e. $r_I(s, a|\theta) := \hat{\mathcal{F}}(\theta)$, where $\hat{\mathcal{F}}(\theta)$ is an estimate of the objective. Note that the right part of the objective can be directly optimized using maximum-entropy RL (Haarnoja et al., 2018; 2017).

Other prominent methods such as DADS and off-DADS (Sharma et al., 2019b; 2020) propose to instead maximize $I(S'; Z|S)$, the mutual information between the next state $s'$ and skill conditioned on the current state. Similar to DIAYN, this objective is maximized by RL optimization of the corresponding intrinsic reward. However, unlike DIAYN, the computation of the reward is model-based, as it requires learning a skill-conditioned dynamics model. In both cases, the $z$-conditioned skill policy $\pi_\theta(\cdot|\cdot; z)$ requires end-to-end joint optimization of all the skills and shares parameters across skills through a single neural network policy.

## 3 METHOD

In DIAYN and DADS, all skills are simultaneously trained from start to end, and they are not designed to be able to adapt to changes in the environment (non-stationary dynamics). This is particularly problematic in lifelong RL settings Nareyek (2003); Koulouriotis & Xanthopoulos (2008); Da Silva et al. (2006); Cheung et al. (2020) or in real-world settings Gupta et al. (2018); Julian et al. (2020) where the environment can change during training.

To address these challenges of reward-free learning and adaptability, we propose incremental policies. Here, instead of simultaneous optimization of skills, they are learned sequentially. This way newer skills can learn to adapt to the evolved environment without forgetting what older skills have learned. In Section 3.1 we discuss our incremental objective, and follow it up with practical considerations and the algorithm in Section 3.2.

## 3.1 Discovery of Incremental Skills (DISk)

To formalize our incremental discovery objective, we begin with Equation 1, under which categorical or discrete skills can be expanded as:

$$\mathcal{F}(\theta) := \mathbb{E}_{z_m \sim p(z)}[IG(S; Z = z_m) + \mathcal{H}(A|S, Z = z_m)] \tag{2}$$

Here, $IG$ refers to the information gain quantity $IG(S; z_m) = \mathcal{H}(S) - \mathcal{H}(S|z_m)$. Intuitively, this corresponds to the reduction in entropy of the state visitation distribution when a specific skill $z_m$ is run. With a uniform skill prior $p(z)$ and $M$ total discrete skills, this expands to:

$$\mathcal{F}(\theta) := \frac{1}{M} \sum_{m=1}^{M} (IG(S; Z = z_m) + \mathcal{H}(A|S, Z = z_m)) \tag{3}$$

In incremental skill discovery, since our goal is to learn a new skill $z_M$ given $M - 1$ previously trained skills $(z_1, z_2, ..., z_{M-1})$ we can fix the learned skills $z_{1:M-1}$, and reduce the objective to:

$$\mathcal{F}(\theta_M) := IG(S; Z = z_M) + \mathcal{H}(A|S, Z = z_M) \tag{4}$$
$$= \mathcal{H}(S) - \mathcal{H}(S|z_M) + \mathcal{H}(A|S, Z = z_M) \tag{5}$$

The first term $\mathcal{H}(S)$ corresponds to maximizing the total entropy of states across all skills, which encourages the new skill $z_M$ to produce diverse behavior different from the previous ones. The second term $-\mathcal{H}(S|z_M)$ corresponds to reducing the entropy of states given the skill $z_M$, which encourages the skill to produce consistent state visitations. Finally, the third term $\mathcal{H}(A|S, Z = z_M)$ encourages maximum entropy policies for the skill $z_M$. Note that this objective can be incrementally applied for $m = 1, \cdots, M$, where $M$ can be arbitrarily large. A step-by-step expansion of this objective is presented in Appendix B.

## 3.2 A Practical Algorithm

Since we treat each skill as an independent policy, our framework involves learning the $M^{th}$ skill $\pi_M$ given a set of prior previously learned skills $\pi_1, \cdots, \pi_{M-1}$. Note that each $\pi_m$ is a stochastic policy representing the corresponding skill $z_m$ and has its own set of learned parameters $\theta_m$. Let $S_m \sim p(s|\pi_m), \overline{S} \sim \sum_{m=1}^{M} p(s|\pi_m)$ denote random variables associated with the states visited by rolling out the policies $\pi_m, 1 \leq m \leq M$. Then, our objective from Equation 5 becomes:

$$\mathcal{F}(\theta_M) = \mathcal{H}(\overline{S}) - \mathcal{H}(S_M) + \mathcal{H}(A|S_M) \tag{6}$$

The rightmost term can be optimized used max-entropy based RL optimization (Haarnoja et al., 2018) as done in prior skill discovery work DIAYN or Off-DADS. However, computing the first two terms is not tractable in large continuous state-spaces. To address this, we employ three practical estimation methods. First, we use Monte-Carlo estimates of the random variables $\overline{S}, S_m$ by rolling out the policy $\pi_m$. Next, we use a point-wise entropy estimator function to estimate the entropy $\mathcal{H}$ from the sampled rollouts. Finally, since distances in raw states are often not meaningful, we measure similarity in a projected space for entropy computation. Given this estimate $\hat{\mathcal{F}}(\theta_M)$ of the objective, we can set the intrinsic reward of the skill as $r_I(s, a|\theta_M) := \hat{\mathcal{F}}(\theta_M)$. The full algorithm is described in Algorithm 1, pseudocode is given in Appendix H, and the key features are described below.

**Monte-Carlo estimation with replay:** For each skill, we collect a reward buffer containing $N$ rollouts from each previous skills that make up our set $\mathcal{T}$. For $\mathcal{T}_M$, we keep a small number of states visited by our policy most recently in a circular buffer such that our $\mathcal{T}_M$ cannot drift too much from the distribution $P_{\pi_M}(s)$. Since point-wise entropy computation of a set of size $n$ has an $O(n^2)$ complexity, we subsample states from our reward buffer to estimate the entropy.

**Point-wise entropy computation:** A direct, Monte-Carlo based entropy estimation method would require us to estimate entropy by computing $-\frac{1}{|S|} \sum_{s \in S} \log p(s)$, which is intractable. Instead, we use a non-parametric, Nearest Neighbor (NN) based entropy estimator (Singh et al., 2003; Liu & Abbeel, 2021; Yarats et al., 2021):

$$\bar{\mathcal{H}}_{\mathbf{X}, k}(p) \propto \frac{1}{|\mathbf{X}|} \sum_{i=1}^{|\mathbf{X}|} \ln ||x_i - \text{NN}_{k, \mathbf{X}}(x_i)||_2 \tag{7}$$

---

**Algorithm 1** Discovery of Incremental Skills: Learning the $M$th Skill

---

**Input:** Projection function $\sigma$, hyperparameter $k, \alpha, \beta$, off-policy learning algorithm $A$.
**Initialize:** Learnable parameters $\theta_M$ for $\pi_M$, empty circular buffer BUF with size $n$, and replay buffer $R$.
Sample trajectories $\mathcal{T} = \bigcup_{m=1}^{M-1}\{s \sim \pi_m\}$ from previous $M - 1$ policies on current environment.
**while** *Not converged* **do**
    Collect transition $(s, a, s')$ by running $\pi_M$ on the environment.
    Store transition data $(s, a, s')$ into the replay buffer, $R \leftarrow (s, a, s')$.
    Add the new projected state to the circular buffer, BUF $\leftarrow \sigma(s')$.
    **for** $t = 1,2,..T_{\text{updates}}$ **do**
        Sample $(s, a, s')$ from $R$.
        Sample batch $\mathcal{T}_b \subset \mathcal{T}$.
        Find $\sigma_c$, the $k^{\text{th}}$-Nearest Neighbor of $\sigma(s')$ within BUF,
        Find $\sigma_d$, the $k^{\text{th}}$-Nearest Neighbor of $\sigma(s')$ within $\sigma(\mathcal{T}_b)$.
        Set consistency penalty $r_c := ||\sigma(s') - \sigma_c||_2$.
        Set diversity reward $r_d := ||\sigma(s') - \sigma_d||_2$.
        Set intrinsic reward $r_I := -\alpha r_c + \beta r_d$.
        Update $\theta_M$ using $A$ using $(s, a, s', r_I)$ as our transition.
**Return:** $\theta_M$

---

where each point $x_i$ of the dataset $\mathbf{X}$ contributes an amount proportional to $\ln ||x_i - \text{NN}_{k,\mathbf{X}}(x_i)||_2$ to the overall entropy. We further approximate $\ln x \approx x - 1$ similar to Yarats et al. (2021) and set $k = 3$ since any small value suffices. Additional details are discussed in Appendix B.

**Measuring similarity under projection:** To measure entropy using Equation 7 on our problem, we have to consider nearest neighbors in an Euclidean space. A good choice of such a space will result in more meaningful diversity in our skills. Hence, in general we estimate the entropy not directly on raw state $s$, but under a projection $\sigma(s)$, where $\sigma$ captures the meaningful variations in our agent. Learning such low-dimensional projections is in itself an active research area (Whitney et al., 2019; Du et al., 2019). Hence, following prior work like Off-DADS and DIAYN that use velocity-based states, we use a pre-defined $\sigma$ that gives us the velocity of the agent. We use instantaneous agent velocity since in locomotion tasks it is a global metric that is generally independent of how long the agent is run for. Like Off-DADS and DIAYN, we found that a good projection is crucial for success of an agent maximizing information theoretic diversity.

**Putting it all together:** On any given environment, we can put together the above three ideas to learn arbitrary number of incremental skills using Algorithm 1. To learn a new skill $\pi_M$ in a potentially evolved environment, we start with collecting an experience buffer $\mathcal{T}$ with states $s \sim \pi_m$ collected from previous skills. Then, we collect transitions $(s, a, s')$ by running $\pi_M$ on the environment. We store this transition in the replay buffer, and also store the projected state $\sigma(s')$ in a circular buffer BUF. Then, to update our policy parameter $\theta_M$, on each update step, we sample $(s, a, s')$ from our replay buffer and calculate the intrinsic reward of that sample. To calculate this, we first find the $k^{\text{th}}$ Nearest Neighbor of $\sigma(s')$ within BUF, called $\sigma_c$ henceforth. Then, we sample a batch $\mathcal{T}_b \subset \mathcal{T}$, and find the $k^{\text{th}}$ Nearest Neighbor of $\sigma(s')$ within $\sigma(\mathcal{T}_b) = \{\sigma(s) \mid s \in \mathcal{T}_b\}$, called $\sigma_d$ henceforth. Given these nearest neighbors, we define our consistency penalty $r_c := ||\sigma(s') - \sigma_c||_2$, and our diversity reward $r_d := ||\sigma(s') - \sigma_d||_2$, which yields an intrinsic reward $r_I := -\alpha r_c + \beta r_d$. $\alpha$ and $\beta$ are estimated such that the expected value of $\alpha r_c$ and $\beta r_d$ are close in magnitude, which is done by using the mean values of $\hat{r}_c$ and $\hat{r}_d$ from the previous skill $\pi_{M-1}$.

Training the first skill with DISk uses a modified reward function, since there is no prior skills $\mathcal{T}$ to compute diversity rewards against. In that case, we simply set $r_I := 1 - \alpha r_c$ throughout the first skills' training, thus encouraging the agent to stay alive while maximizing the consistency. Exact hyperparameter settings are provided in Appendix D.

## 4 EXPERIMENTS

We have presented an unsupervised skill discovery method DISk that can learn decoupled skills incrementally without task-specific rewards. In this section we answer the following questions: (a)

Does DISk learn diverse and consistent skills in evolving environments with non-stationary dynamics? (b) How does DISk compare against traditional skill learning methods in stationary environments? (c) Can skills learned by DISk accelerate the learning of downstream tasks? (d) How does our specific design choices effect the performance of DISk? To answer these questions we first present our experimental framework and baselines, and then follow it with our experimental results.

## 4.1 Experimental setup

To study DISk, we train skills on an agent that only has access to the environment without information about the downstream tasks (Eysenbach et al., 2018; Sharma et al., 2019b; 2020). We use two types of environments: for stationary tasks we use standard MuJoCo environments from OpenAI Gym (Todorov et al., 2012; Brockman et al., 2016): HalfCheetah, Hopper, Ant, and Swimmer, visualized in Fig. 4 (left), while for non-stationary environments we use two modified variants of the Ant environment: one with disappearing blocks, and another with broken legs. Environment details are provided in Appendix C.

**Baselines:** In our experiments, we compare our algorithm against two previous state-of-the-art unsupervised skill discovery algorithms mentioned in Section 2, DIAYN (Eysenbach et al., 2018) and Off-DADS (Sharma et al., 2020). To contextualize the performance of these algorithms, whenever appropriate, we compare them with a collection of random, untrained policies. To make our comparison fair, we give both DIAYN and Off-DADS agents access to the same transformed state, $\sigma(s)$, that we use in DISk. Since Off-DADS is conditioned by continuous latents, for concrete evaluations we follow the same method as the original paper by uniformly sampling $N$ latents from the latent space, and using them as the $N$ skills from Off-DADS in all our experiments. Exact hyperparameter settings are described in Appendix D, and our implementation of DISk is attached as supplimentary material.

## 4.2 Can DISk adaptively discover skills in evolving environments?

One of the key hypothesis of this work is that DISk is able to discover useful skills even when the environment evolves over time. To gauge the extent of DISk's adaptability, we design two experiments shown in Fig. 2 (left) and Fig. 1 (b) for judging adaptability of DISk in environments that change in continuous and discrete manner respectively.

**Continuous Change:** We create a MuJoCo environment similar to Ant with 40 blocks encircling the agent. The blocks initially prevent long trajectories in any direction. Then, as we train the agent, we slowly remove the blocks at three different speeds. If $T$ is the total training time of the agent, in "fast", we remove one block per $T/40$ steps, in "even" we remove two blocks per $T/20$ training steps, and in "slow" we remove ten blocks per $T/4$ training steps. In each of the cases, the environment starts with a completely encircled agent, and ends with an agent completely free to move in any direction; across the three environments just the frequency and the magnitude of the evolution changes. This environment is meant to emulate an expanding environment where new possibilities open up during training; like a domestic robot being given access to new rooms in the house. The three different rates of change lets us observe how robust a constant skill addition schedule is with different rates of change in the world.

In Fig. 2 (middle), we can see how the skill learning algorithms react to the changing environment. The comparison between Off-DADS and DISk is especially noticeable. The Off-DADS agent latches onto the first opening of the block-circle, optimizes the skill dynamics model for motions on that single direction, and ignores further evolution of the environment quite completely. On the other hand, since the DISk agent incrementally initializes an independent policy per skill, it can easily adapt to the new, changed environment during the training time of each skill. To quantitatively measure the diversity of skills learned, we use the Hausdorff distance (Belogay et al., 1997), which is a topological way of measuring distance between two sets of points (see appendix B.3.) We plot the mean Hausdorff distance between all endpoints of a skill and all other endpoints in Fig. 2 (right). In this metric as well, DISk does better than other methods across environments.

**Discrete Change:** We modify the standard Ant environment to a changing dynamic environment by disabling one of the four Ant legs during test and train time. To make the environment dynamic, we cycle between broken legs every 1M steps during the 10M steps of training. We show the trajectories of the learned skills with each of the legs broken in Fig. 3.

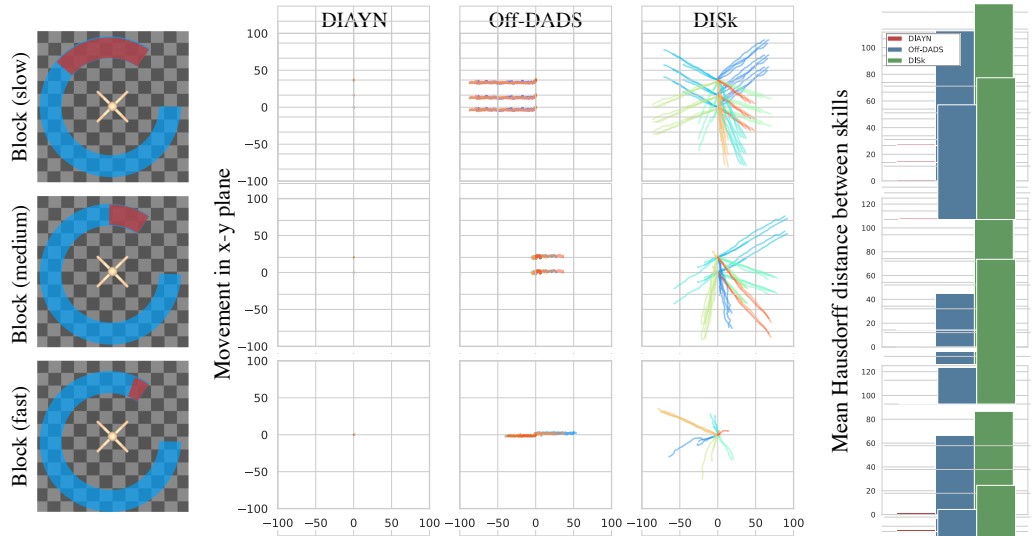

Figure 2: Left: Block environments that evolve during training; red highlights portion of blocks vanishing at once. Middle: Trajectories of skills discovered by different algorithms; each unique color is a different skill. Right: Mean Hausdorff distance of the skills discovered in these environments.

In this case, the Off-DADS and the DISk agent both adapt to the broken leg and learn some skills that travel a certain distance over time. However, a big difference becomes apparent if we note under which broken leg the Off-DADS agent is performing the best. The Off-DADS agent, as seen in Fig. 3 (left), performs well with the broken leg that it was trained on most recently. To optimize for this recently-broken leg, this agent forgets previously learned skills (App. Fig. 7, 8). Conversely, each DISk skill learns how to navigate with a particular leg broken, and once a skill has learned to navigate with some leg broken, that skill always performs well with that particular dynamic.

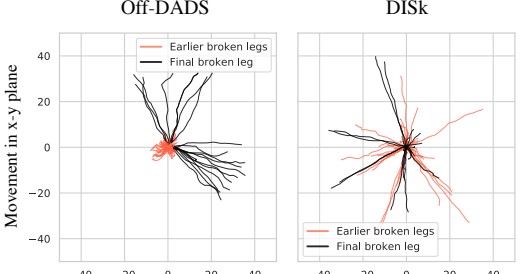

Figure 3: Evaluation of skills learned on the broken Ant environment with different legs of the Ant disabled; leg disabled during evaluation in legend.

### 4.3 WHAT QUALITY OF SKILLS ARE LEARNED BY DISk IN A STATIC ENVIRONMENT?

While DISk was created to address skill learning in an evolving environment, intuitively it could work just as well in a static environment. To qualitatively study DISk-discovered skills in static environments, we plot the trajectories generated by them in Fig. 4 (middle). On the three hardest environments (Hopper, Ant, and Swimmer), DISk produces skills that cover larger distances than DIAYN and Off-DADS. Moreover, on Ant and Swimmer, we can see that skills from DISk are quite consistent with tight packing of trajectories. On the easier HalfCheetah environment, we notice that DIAYN produces higher quality skills. One reason for this is that both DIAYN and Off-DADS share parameters across skills, which accelerates skill learning for less complex environments. Also, note that the performance of DIAYN is worse on the complex environments like Swimmer and Ant. The DIAYN agent stops learning when the DIAYN discriminator achieves perfect discriminability; since we are providing the agent and the discriminator with extra information $\sigma(s)$, in complex environments like Swimmer or Ant DIAYN achieves this perfect discriminability and then stops learning. While the performance of DIAYN on Ant may be surprising, it is in line with the observations by Sharma et al. (2019b; 2020).

As shown in the quantitative results in Fig. 4 (right), for the harder Hopper, Ant, and Swimmer tasks, DISk demonstrates substantially better performance. However, on the easier HalfCheetah task,

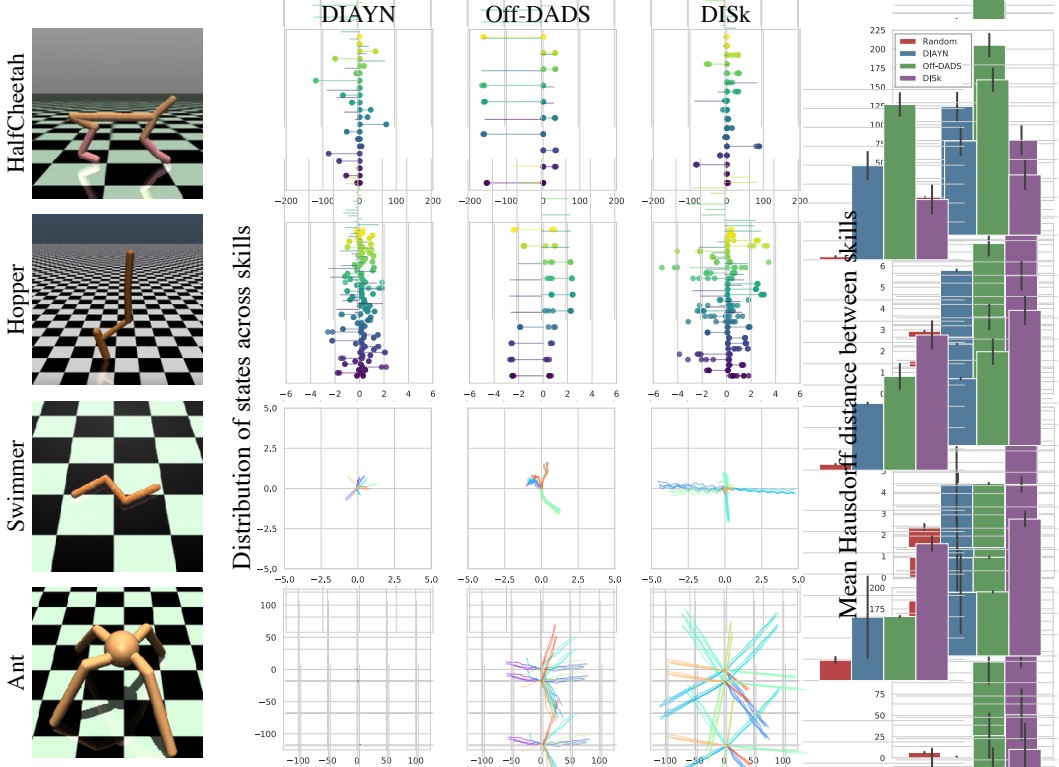

Figure 4: Left: Environments we evaluate on. Middle: Visualization of trajectories generated by the skills learned by DISk and baselines, with each unique color denoting a specific skill. For environments that primarily move in the x-axis (HalfCheetah and Hopper), we plot the final state reached by the skill across five runs. For environments that primarily move in the x-y plane (Ant and Swimmer), we plot five trajectories for each skill. Right: Hausdorff distance (higher is better) between the terminal states visited by each skill and terminal states of all other skills, averaged across the set of discovered skills to give a quantitative estimate of the skill diversity.

DISk underperforms DIAYN and Off-DADS. This supports our hypothesis that incremental learning discovers higher quality skills in hard tasks, while prior works are better on easier environments.

### 4.4 CAN DISk SKILLS ACCELERATE DOWNSTREAM LEARNING?

A promise of unsupervised skill discovery methods is that in complicated environments the discovered skills can accelerate learning on downstream tasks. In our experiments, we find that skills learned by DISk can learn downstream tasks faster by leveraging the learned skills hierarchically. To examine the potential advantage of using discovered skills, we set up an experiment similar to the goal-conditioned hierarchical learning experiment seen in Sharma et al. (2019b). In this experiment, we initialize a hierarchical Ant agent with the learned skills and task it with reaching an $(x, y)$ coordinate chosen uniformly from $[-15, 15]^2$. Note that this environment is the standard Ant environment with no changing dynamics present during evaluations. To highlight the versatility of the skills learned under evolving environments, we also evaluate the agents that were trained under our evolving Block environments (see Section 4.2) under the same framework.

We generally find that DISk agents outperform other agents on this hierarchical learning benchmark, by generally converging within 100-200k steps compared to DIAYN agent never converging and Off-DADS taking around 500k steps (Fig. 5). This statement holds true regardless of the evolution dynamics of the environment where the agents were learned, thus showing that DISk provides an adaptive skill discovery algorithm under a variety of learning circumstances. This property makes DISk particularly suitable for lifelong learning scenarios where the environment is continuously evolving.

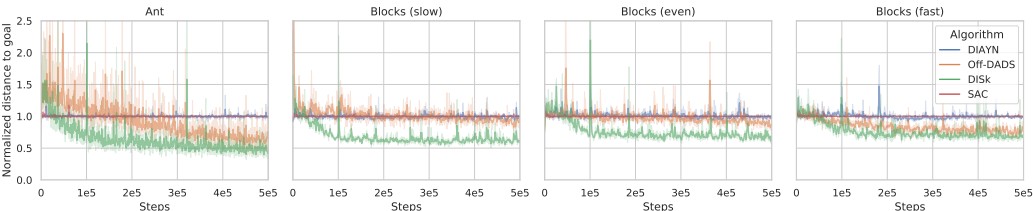

Figure 5: Downstream Hierarchical Learning. We plot the average normalized distance (lower is better), given by $\frac{1}{T}\sum_{t=0}^{T}(d(s_t, g)/d(s_0, g))$ from our Ant agent to the target over 500K training steps. From left to right, we plot the performance of agents that were trained in environment with no obstacles, and our three evolving block environments. Shading shows variance over three runs, and we add a goal-based Soft Actor Critic model trained non-hierarchically as a baseline. Across all settings, DISk outperforms prior work.

## 4.5 ABLATION ANALYSIS ON DESIGN CHOICES

We make several design choices that improve DISk. In this section we briefly discuss the two most important ones. Firstly, our implementation of DISk shares the data collected by the previous skills by initializing the replay buffer $\mathcal{R}$ of a new skill with data from the previous ones. Once initialized with this replay, the new skill can relabel that experience with its own intrinsic rewards. This method has shown promise in multi-task RL and hindsight relabelling (Andrychowicz et al., 2017; Li et al., 2020; Eysenbach et al., 2020). Across our experiments, performing such relabelling improves performance significantly for the later skills (App. Fig. 10 (left)) as it can reuse vast amounts of data seen by prior skills. Training details for this experiment is presented in Appendix G.

Compared to prior work, DISk not only learns skills incrementally, but also learns independent neural network policies for each skill. To check if the performance gains in static environment are primarily due to the incremental addition of skills or the independent policy part, we run a version of our algorithm where all the independent skills are trained in parallel (App. Fig. 10 (left)) on Ant. Experimentally, we see that simply training independent skills in parallel does not discover useful skills; with the parallel variant achieving far lower mean Hausdorff distance. Note that we cannot have incremental addition of skills without independent policies, at least in the current instantiation of DISk, without significant modification to the algorithm or the architecture. This is because it is highly nontrivial to extend or modify a network's input or parameter space, and the simplest extension is just a new network which is DISk itself.

## 5 RELATED WORK

Our work on developing incremental skill discovery is related to several subfields of AI. In this section we briefly describe the most relevant ones.

**Incremental Learning:** Machine learning algorithms have traditionally focused on stationary, non-incremental learning, where a fixed dataset is presented to the algorithm (Giraud-Carrier, 2000; Sugiyama & Kawanabe, 2012; Ditzler et al., 2015; Lee et al., 2018; 2017; Smith & Gasser, 2005). Inspired by studies in child development (Mendelson et al., 1976; Smith & Gasser, 2005; Smith et al., 1998; Spelke, 1979; Wertheimer, 1961), sub-areas such as curriculum learning (Bengio et al., 2009; Kumar et al., 2010; Murali et al., 2018) and incremental SVMs (Cauwenberghs & Poggio, 2001; Syed et al., 1999; Ruping, 2001) have emerged, where learning occurs incrementally. We note that the general focus in these prior work is on problems where a labelled dataset is available. In this work, we instead focus on RL problems, where labelled datasets or demonstrations are not provided during the incremental learning. In the context of RL, some prior works consider learning for a sequence of tasks (Tanaka & Yamamura, 1997; Wilson et al., 2007; Ammar et al., 2015) while other works (Peng & Williams, 1994; Ammar et al., 2015; Wang et al., 2019a;b) create algorithms that can adapt to changing environments through incremental policy or Q-learning. However, these methods operate in the task-specific RL setting, where the agent is trained to solve one specific reward function.

**Unsupervised Learning in RL:** Prominent works in this area focus on obtaining additional information about the environment in the expectation that it would help downstream task-based RL: Representation learning methods (Yarats et al., 2019; Lee et al., 2019; Srinivas et al., 2020; Schwarzer

et al., 2020; Stooke et al., 2020; Yarats et al., 2021) use unsupervised perceptual losses to enable better visual RL; Model-based methods (Hafner et al., 2018; 2019; Yan et al., 2020; Agrawal et al., 2016) learn approximate dynamics models that allow for accelerated planning with downstream rewards; Exploration-based methods (Bellemare et al., 2016; Ostrovski et al., 2017; Pathak et al., 2017; Burda et al., 2018; Andrychowicz et al., 2017; Hazan et al., 2019; Mutti et al., 2020; Liu & Abbeel, 2021) focus on obtaining sufficient coverage in environments. Since our proposed work is based on skill discovery, it is orthogonal to these prior work and can potentially be combined with them for additional performance gain. Most related to our work is the sub-field of skill discovery (Eysenbach et al., 2018; Sharma et al., 2019b; Campos et al., 2020; Gregor et al., 2016; Achiam et al., 2018; Sharma et al., 2020). A more formal treatment of this is presented in Section 2. Since our method makes the skill discovery process incremental, it allows for improved performance compared to DIAYN (Eysenbach et al., 2018) and Off-DADS (Sharma et al., 2020), which is discussed in Section 4. Finally, we note that skills can also be learned in supervised setting with demonstrations or rewards (Kober & Peters, 2009; Peters et al., 2013; Schaal et al., 2005; Dragan et al., 2015; Konidaris et al., 2012; Konidaris & Barto, 2009; Zahavy et al., 2021; Shankar et al., 2020; Shankar & Gupta, 2020). Our work can potentially be combined with such prior work to improve skill discovery when demonstrations are present.

**RL with evolving dynamics:** Creating RL algorithms that can adapt and learn in environments with changing dynamics is a longstanding problem (Nareyek, 2003; Koulouriotis & Xanthopoulos, 2008; Da Silva et al., 2006; Cheung et al., 2020). Recently, a variant of this problem has been studied in the sim-to-real transfer community (Peng et al., 2018; Chebotar et al., 2019; Tan et al., 2018), where minor domain gaps between simulation and reality can be bridged through domain randomization. Another promising direction is online adaptation (Rakelly et al., 2019; Hansen et al., 2020; Yu et al., 2017) and Meta-RL (Finn et al., 2017; Nagabandi et al., 2018a;b), where explicit mechanisms to infer the environment are embedded in policy execution. We note that these methods often focus on minor variations in dynamics during training or assume that there is an underlying distribution of environment variability, while our framework does not.

## 6 DISCUSSION, LIMITATIONS, AND SCOPE FOR FUTURE WORK

We have presented DISk, an unsupervised skill discovery method that takes the first steps towards learning skills incrementally. Although already powerful, we still have significant scope for future work before such methods can be applied in real-world settings. For instance, we notice that on easy tasks such as HalfCheetah, DISk underforms prior works, which is partly due to its inability to share parameters from previously learned skills. Sharing parameters efficiently can allow improved learning and a reduction of the total parameters. Next, to measure similarity in states, we use a fixed projection function. Being able to simultaneously learn this projection function is likely to further improve performance. Also, we use a fixed schedule based on skill convergence to add new skills to our repertoire. A more intelligent schedule for skill addition would help DISk be even more optimized. Finally, to apply such skill discovery methods in the real-world, we may have to bootstrap from supervised data or demonstrations. Such bootstrapping can provide an enormous boost in settings where humans can interact with the agent.

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

## APPENDIX

## A  REINFORCEMENT LEARNING

In our continuous-control RL setting, an agent receives a state observation $s_t \in \mathcal{S}$ from the environment and applies an action $a_t \in \mathcal{A}$ according to policy $\pi$. In our setting, where the policy is stochastic, the policy returns a distribution $\pi(s_t)$, and we sample a concrete action $a_t \sim \pi(s_t)$. The environment returns a reward for every action $r_t$. The goal of the agent is to maximize expected cumulative discounted reward $\mathbb{E}_{s_{0:T}, a_{0:T-1}, r_{0:T-1}} \left[ \sum_{t=0}^{T-1} \gamma^t r_t \right]$ for discount factor $\gamma$ and horizon length $T$.

On-policy RL (Schulman et al., 2015; Kakade, 2002; Williams, 1992) optimizes $\pi$ by iterating between data collection and policy updates. It hence requires new on-policy data every iteration, which is expensive to obtain. On the other hand, off-policy reinforcement learning retains past experiences in a replay buffer and is able to re-use past samples. Thus, in practice, off-policy algorithms have been found to achieve better sample efficiency (Lillicrap et al., 2015; Haarnoja et al., 2018). For our experiments we use SAC (Haarnoja et al., 2018) as our base RL optimizer due to its implicit maximization of action distribution entropy, sample efficiency, and fair comparisons with baselines that also build on top of SAC. However, we note that our framework is compatible with any standard off-policy RL algorithm that maximizes the entropy of the action distribution $\pi(\cdot)$ either implicitly or explicitly.

**Soft Actor-Critic**   The Soft Actor-Critic (SAC) (Haarnoja et al., 2018) is an off-policy model-free RL algorithm that instantiates an actor-critic framework by learning a state-action value function $Q_\theta$, a stochastic policy $\pi_\theta$ and a temperature $\alpha$ over a discounted infinite-horizon MDP $(\mathcal{X}, \mathcal{A}, P, R, \gamma, d_0)$ by optimizing a $\gamma$-discounted maximum-entropy objective (Ziebart et al., 2008). With a slight abuse of notation, we define both the actor and critic learnable parameters by $\theta$. SAC parametrizes the actor policy $\pi_\theta(\boldsymbol{a}_t|\boldsymbol{x}_t)$ via a $\tanh$-Gaussian defined as $\boldsymbol{a}_t = \tanh(\mu_\theta(\boldsymbol{x}_t) + \sigma_\theta(\boldsymbol{x}_t)\epsilon)$, where $\epsilon \sim \mathcal{N}(0,1)$, $\mu_\theta$ and $\sigma_\theta$ are parametric mean and standard deviation. The SAC's critic $Q_\theta(\boldsymbol{x}_t, \boldsymbol{a}_t)$ is parametrized as an MLP neural network.

The policy evaluation step learns the critic $Q_\theta(\boldsymbol{x}_t, \boldsymbol{a}_t)$ network by optimizing the one-step soft Bellman residual:

$$\mathcal{L}_Q(\mathcal{D}) = \mathbb{E}_{\substack{(\boldsymbol{x}_t, \boldsymbol{a}_t, \boldsymbol{x}_{t+1}) \sim \mathcal{D} \\ \boldsymbol{a}_{t+1} \sim \pi(\cdot|\boldsymbol{x}_{t+1})}}[(Q_\theta(\boldsymbol{x}_t, \boldsymbol{a}_t) - y_t)^2] \text{ and}$$
$$y_t = R(\boldsymbol{x}_t, \boldsymbol{a}_t) + \gamma[Q_{\theta'}(\boldsymbol{x}_{t+1}, \boldsymbol{a}_{t+1}) - \alpha \log \pi_\theta(\boldsymbol{a}_{t+1}|\boldsymbol{x}_{t+1})],$$

where $\mathcal{D}$ is a replay buffer of transitions, $\theta'$ is an exponential moving average of $\theta$ as done in (Lillicrap et al., 2015). SAC uses clipped double-Q learning (Van Hasselt et al., 2016; Fujimoto et al., 2018), which we omit from our notation for simplicity but employ in practice.

The policy improvement step then fits the actor $\pi_\theta(\boldsymbol{a}_t|\boldsymbol{s}_t)$ network by optimizing the following objective:

$$\mathcal{L}_\pi(\mathcal{D}) = \mathbb{E}_{\boldsymbol{x}_t \sim \mathcal{D}}[D_{\mathrm{KL}}(\pi_\theta(\cdot|\boldsymbol{x}_t) || \exp\{\frac{1}{\alpha} Q_\theta(\boldsymbol{x}_t, \cdot)\})].$$

Finally, the temperature $\alpha$ is learned with the loss:

$$\mathcal{L}_\alpha(\mathcal{D}) = \mathbb{E}_{\substack{\boldsymbol{x}_t \sim \mathcal{D} \\ \boldsymbol{a}_t \sim \pi_\theta(\cdot|\boldsymbol{x}_t)}}[-\alpha \log \pi_\theta(\boldsymbol{a}_t|\boldsymbol{x}_t) - \alpha\bar{\mathcal{H}}],$$

where $\bar{\mathcal{H}} \in \mathbb{R}$ is the target entropy hyper-parameter that the policy tries to match, which in practice is set to $\bar{\mathcal{H}} = -|\mathcal{A}|$. The overall optimization objective of SAC equals to:

$$\mathcal{L}_{\mathrm{SAC}}(\mathcal{D}) = \mathcal{L}_\pi(\mathcal{D}) + \mathcal{L}_Q(\mathcal{D}) + \mathcal{L}_\alpha(\mathcal{D}).$$

# B   FURTHER MATHEMATICAL DETAILS

## B.1   EXPANSION AND DERIVATION OF OBJECTIVES

The objective function $\mathcal{F}(\theta)$, as defined in Equation 1, is the source from where we derive our incremental objective function, reproduced here.

$$\mathcal{F}(\theta) = I(S; Z) + \mathcal{H}(A|S, Z)$$

We can expand the first term in Equation 1 as

$$I(S; Z) \equiv H(S) - H(S \mid Z)$$

by definition of mutual information. Now, once we assume $Z$ is a discrete variable, the second part of this equation becomes

$$\mathcal{H}(S \mid Z) \equiv \sum_{z_m} p(z_m) \mathcal{H}(S \mid Z = z_m)$$
$$= \mathbb{E}_{z_m \sim p(z_m)} [\mathcal{H}(S \mid Z = z_m)]$$

And thus we have

$$I(S; Z) = \mathcal{H}(S) - \mathbb{E}_{z_m \sim p(z_m)} [\mathcal{H}(S \mid Z = z_m)]$$
$$= \mathbb{E}_{z_m \sim p(z_m)} [\mathcal{H}(S) - \mathcal{H}(S \mid Z = z_m)]$$

But the term inside the expectation is the definition of information gain (not to be confused with KL divergence), defined by

$$IG(S; Z = z_m) \equiv \mathcal{H}(S) - \mathcal{H}(S \mid Z = z_m)$$

Thus, we arrive at

$$I(S; Z) = \mathbb{E}_{z_m \sim p(z_m)} [IG(S; Z = z_m)]$$

Similarly, by definition of conditional entropy, we can expand the second part of the Equation 1

$$\mathcal{H}(A \mid S, Z) \equiv \sum_{z_m} p(z_m) \mathcal{H}(A \mid S, Z = z_m)$$
$$= \mathbb{E}_{z_m \sim p(z_m)} [\mathcal{H}(A \mid S, Z = z_m)]$$

Thus, we can convert Equation 1 into

$$\mathcal{F}(\theta) = I(S; Z) + \mathcal{H}(A \mid S, Z)$$
$$= \mathbb{E}_{z_m \sim p(z_m)} [IG(S; Z = z_m)] + \mathbb{E}_{z_m \sim p(z_m)} [\mathcal{H}(A \mid S, Z = z_m)]$$
$$= \mathbb{E}_{z_m \sim p(z_m)} [IG(S; Z = z_m) + \mathcal{H}(A \mid S, Z = z_m)]$$

If we assume a uniform prior over our skills, which is another assumption made by Eysenbach et al. (2018), and also assume we are trying to learn $M$ skills in total, we can further expand Equation 1 into:

$$\mathcal{F}(\theta) = \frac{1}{M} \sum_{m=1}^{M} [IG(S; Z = z_m) + \mathcal{H}(A \mid S, Z = z_m)]$$

Ignoring the number of skills term (which is constant over a single skills' learning period) gives us exactly Equation 3, which was:

$$\mathcal{F}(\theta) := \sum_{m=1}^{M} [IG(S; Z = z_m) + \mathcal{H}(A \mid S, Z = z_m)]$$

Now, under our framework, we assume that skills $1, 2, \cdots, M - 1$ has been learned and fixed, and we are formulating an objective for the $M$th skill. As a result, we can ignore the associated Information Gain and action distribution entropy terms from skills $1, 2, \cdots, M - 1$, and simplify our objective to be:

$$\mathcal{F}(\theta) := IG(S; Z = z_M) + \mathcal{H}(A \mid S, Z = z_M)$$
$$= \mathcal{H}(S) - \mathcal{H}(S \mid Z = z_M) + \mathcal{H}(A \mid S, Z = z_M)$$

which is exactly the same objective we defined in Equation 5.

## B.2   POINT-BASED NEAREST NEIGHBOR ENTROPY ESTIMATION

In our work, we use an alternate approach, first shown by Singh et al. (2003), to estimate the entropy of a set of points. This method gives us a non-parametric Nearest Neighbor (NN) based entropy estimator:

$$\hat{\mathbb{H}}_{k,\boldsymbol{X}}(p) = -\frac{1}{N}\sum_{i=1}^{N}\ln\frac{k\Gamma(q/2+1)}{N\pi^{q/2}R_{i,k,\boldsymbol{X}}^{q}} + C_k,$$

where $\Gamma$ is the gamma function, $C_k = \ln k - \frac{\Gamma'(k)}{\Gamma(k)}$ is the bias correction term, and $R_{i,k,\boldsymbol{X}} = \|\boldsymbol{x}_i - \mathrm{NN}_{k,\boldsymbol{X}}(\boldsymbol{x}_i)\|$ is the Euclidean distance between $\boldsymbol{x}_i$ and its $k^{\text{th}}$ nearest neighbor from the dataset $\boldsymbol{X}$, defined as $\mathrm{NN}_{k,\boldsymbol{X}}(\boldsymbol{x}_i)$.

The term inside the sum can be simplified as

$$\begin{aligned}
\ln\frac{k\Gamma(q/2+1)}{N\pi^{q/2}R_{i,k,\boldsymbol{X}}^{q}} &= \ln\frac{k\Gamma(q/2+1)}{N\pi^{q/2}} - \ln R_{i,k,\boldsymbol{X}}^{q}\\
&= \ln\frac{k\Gamma(q/2+1)}{N\pi^{q/2}} - q\ln R_{i,k,\boldsymbol{X}}\\
&= \ln\frac{k\Gamma(q/2+1)}{N\pi^{q/2}} - q\ln\|\boldsymbol{x}_i - \mathrm{NN}_{k,\boldsymbol{X}}(\boldsymbol{x}_i)\|.
\end{aligned}$$

Here, $\ln\frac{k\Gamma(q/2+1)}{N\pi^{q/2}}$ is a constant term independent of $\boldsymbol{x}_i$. If we ignore the this term and the bias-correction term $C_k$ and the constant, we get

$$\hat{\mathbb{H}}_{k,\boldsymbol{X}}(p) \propto \sum_{i=1}^{N}\ln\|\boldsymbol{x}_i - \mathrm{NN}_{k,\boldsymbol{X}}(\boldsymbol{x}_i)\|.$$

Which is the formulation we use in this work. This estimator is shown to be asymptotically unbiased and consistent in Singh et al. (2003).

## B.3   HAUSDORFF DISTANCE

In our work, to compare between two algorithms learning skills on the same environment, we used a metric based on Hausdorff distance. Hausdorff distance, also known as the Hausdorff metric or the Pompeiu–Hausdorff distance, is a metric that measures the distance between two subsets of a metric space. Informally, we think of two sets in a metric space as close in the Hausdorff distance if every point of either set is close to some point of the other set. The Hausdorff distance is the longest distance one can force you to travel by choosing a point adversarially in one of the two sets, from which you have to travel to the other set. Put simply, it is the greatest of all the distances from a point in one set to the nearest point in the other.

Mathematically, given two subsets $A$ and $B$ of a metric space $(M, d)$ we define Hausdorff distance $d_H(A, B)$ as:

$$d_H(A, B) = \max\left\{\sup_{a\in A}d(a, B), \sup_{b\in B}d(b, A)\right\}$$

Where $\sup$ represents the supremum, $d(x, Y) = \inf_{y\in Y}d(x, y)$ is the distance between a point and another set, and $\inf$ represents the infimum.

Given a set of skills, we calculate the diversity of one skill over all other skills by calculating the Hausdorff distance between that skill's trajectory end $(x, y)$ location, and the terminal $(x, y)$ locations of all other trajectories. Intuitively, a skill has high Hausdorff distance if the end states it generates is far away from other skills' endpoints. Similarly, a high average Hausdorff distance for skills from an algorithm means that the algorithm's generated skills on average have a high distance from each other, which is a desirable property for an algorithm which needs to generate diverse skills.

## C    Environments

**Gym Experiments**    We derive all environments used in the experiments in this paper from OpenAI Gym (Brockman et al., 2016) MuJoCo tasks. Namely, we use the HalfCheetah-v3 and Hopper-v3 environments for 2d locomotion tasks and Swimmer-v3 and Ant-v3 environments for the 3d locomotion tasks (see Figure 2 for the agent morphologies).

Since we aim to train primitives, we want policies that perform well regardless of the global states of the agent (global position etc.), only depending on local states (join angles etc.). Thus, we train our agents and each of our baselines with a maximum episode length of 100 (200 for Swimmer only), while we test them with a maximum episode length of 500 for static or the block environments and 200 for the broken leg environments.

As our projection function $\sigma$, we measured the $x$ velocity of the agent in 2d environments, and the $(x, y)$ velocity of the agent in 3d environments. We made $\sigma(s)$ available to the intrinsic reward calculation functions of both our methods and the baselines.

**Block Experiments**    For our block experiment set, we implemented the blocks as immovable spheres of radius 3 at a distance 10 from origin. We dynamically added 40 blocks at the environment creation, and deleted them with the MuJoCo interface available in Gym. The blocks were all added before the agent took the first step in the environment, and removed over the agents' lifetime as described in Section 4.3. The blocks were always removed counter-clockwise, following the trajectory of $(\cos\frac{2\pi t}{T}, \sin\frac{2\pi t}{T})$ over $t \in [0, T]$, where $t$ is the current timestep and $T$ is the total timestep for training.

**Broken Leg Experiments**    For our broken leg experiment set, we implemented a broken leg as a leg where no actions have any effect. We switch which leg is broken every 1M steps, and train all skills for a total of 10M steps in both Off-DADS and DISk.

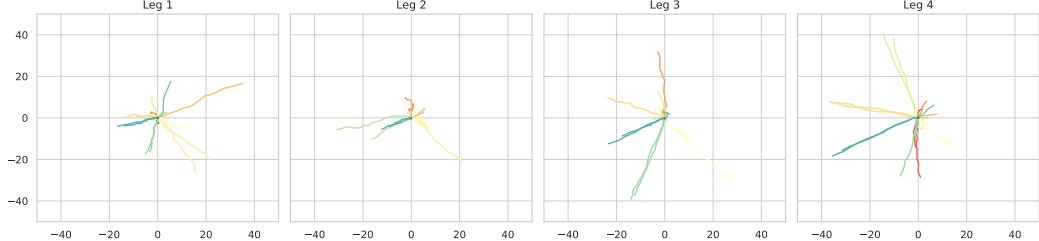

Figure 6: Skills learned by DISk, evaluated with each of the legs broken. The legs are numbered such that the final leg is numbered #4

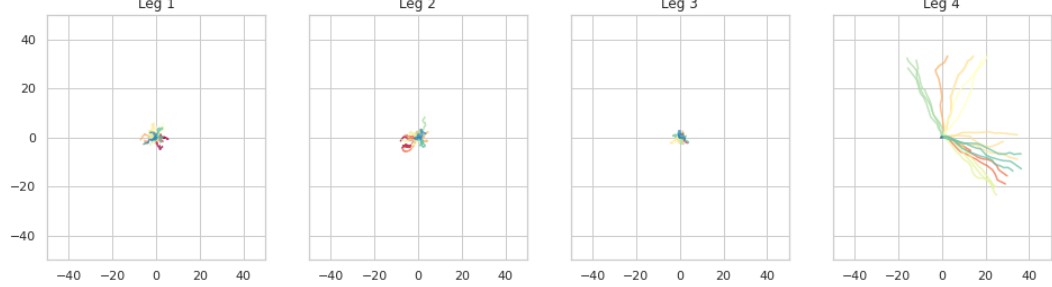

Figure 7: Skills learned by Off-DADS at the end of 10M steps, evaluated with each of the legs broken. The legs are numbered such that the final leg is numbered #4,

**Hierarchical Experiments**    For the hierarchical environments, we use the Ant-v3 environment in a goal-conditioned manner. The goals are sampled from $[-15, 15]^2$ uniformly, and the hierarchical agent can take 100 steps to reach as close to the goal as possible. At each step of the hierarchical

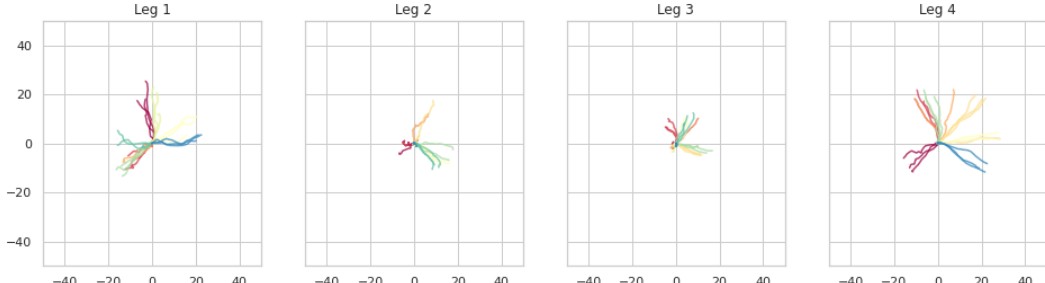

Figure 8: Skills learned by Off-DADS at the end of 9.3M, or 0.3M steps after breaking the final leg, evaluated with each of the legs broken. Compared to this agent, the agent in Fig. 7 performs worse on all broken leg but the last one, which shows that Off-DADS suffers from an instance of catastrophic forgetting.

agent, it chooses a skill, which is then executed for 10 timesteps in the underlying environment. So, in total, the agent has 1000 timesteps to reach the goal. At every timestep, the agent is given a dense reward of $-\|x - g\|_2$, where $x$ is the current location of the agent, and $g$ is the location of the goal. On each step, the hierarchical agent gets the sum of the goal conditioned reward from the 10 timesteps in the underlying environment.

All the hierarchical agents were trained with the `stable-baselines3` package (Raffin et al., 2019). We used their default PPO agent for all the downstream set of skills, and trained the hierarchical agent for a total 500 000 environment steps.

## D  IMPLEMENTATION DETAILS AND HYPERPARAMETERS

We base our implementation on the PyTorch implementation of SAC by Yarats & Kostrikov (2020). On top of the implementation, we add a reward buffer class that keeps track of the intrinsic reward. We provide the pseudocode for our algorithm in Appendix H.

### D.1  ARCHITECTURE

For all of our environments, we used MLPs with two hidden layers of width 256 as our actor and critic networks. We used ReLU units as our nonlinearities. For our stochastic policies, the actor networks generated a mean and variance value for each coordinate of the action, and to sample an action we sampled each dimension from a Gaussian distribution defined by those mean and variance values. For stability, we clipped the log standard deviation between $[-5, 2]$, similar to Yarats & Kostrikov (2020).

### D.2  REWARD NORMALIZATION

In learning all skills except the first, we set $\alpha$ and $\beta$ such that both the diversity reward and the consistency penalty have a similar magnitude. We do so by keeping track of the average of the previous skills' diversity reward and consistency penalty, and using the inverse of that as $\beta$ and $\alpha$ respectively, which is largely successful at keeping the two terms to similar orders of magnitude.

Another trick that helps the DISk learn is using a tanh-shaped annealing curve to scale up $\alpha$, which starts at 0 and ends at its full value. This annealing is designed to keep only the diversity reward term relevant at the beginning of training, and then slowly introduce a consistency constraint into the objective over the training period. This annealing encourages the policy to explore more in the beginning. Then, as the consistency term kicks in, the skill becomes more and more consistent while still being diverse. As a note, while DISk is more efficient with this annealing, its success is not dependent on it.

Table 1: DISks list of hyper-parameters.

| Parameter | Setting |
|---|---|
| Replay buffer capacity (static env) | 2 000 000 |
| Replay buffer capacity (changing env) | 4 000 000 |
| Seed steps | 5000 |
| Per-skill collected steps | 10 000 |
| Minibatch size | 256 |
| Discount ($\gamma$) | 0.99 |
| Optimizer | Adam |
| Learning rate | $3 \times 10^{-4}$ |
| Critic target update frequency | 2 |
| Critic target EMA momentum ($\tau_{\mathrm{Q}}$) | 0.01 |
| Actor update frequency | 2 |
| Actor log stddev bounds | $[-5, 2]$ |
| Encoder target update frequency | 2 |
| Encoder target EMA momentum ($\tau_{\mathrm{enc}}$) | 0.05 |
| SAC entropy temperature | 0.1 |
| Number of samples in $\mathcal{T}_b$ | 256 |
| Size of circular buffer BUF | 50 |
| $k$ in NN | 3 |

Table 2: DISks number of steps per skill.

| Environment | (Average) steps per skill | Number of skills | Total number of steps |
|---|---|---|---|
| HalfCheetah-v3 | 125 000 | 20 | 2 500 000 |
| Hopper-v3 | 50 000 | 50 | 2 500 000 |
| Swimmer-v3 | 250 000 | 10 | 2 500 000 |
| Ant-v3 | 500 000 | 10 | 5 000 000 |
| Ant-v3 (blocks) | 500 000 | 10 | 5 000 000 |
| Ant-v3 (broken) | 500 000 | 20 | 10 000 000 |

## D.3    FULL LIST OF HYPER-PARAMETERS

## D.4    LEARNING SCHEDULE FOR ANT-V3 ON DISK

We noticed that on the static Ant environment, not all learned skills have the same complexity. For example, the first two skills learned how to sit in place, and flip over. On the other hand, later skills which just learns how to move in a different direction could reuse much of the replay buffers of the earlier, complex skills which learned to move, and thus they did not need as many training steps as the earlier complex skills. As a result, they converged at different speed. A schedule that most closely fits this uneven convergence time is what we used: 5 000 000 environment steps unevenly between the skills, with the skills getting 250 000, 250 000, 1 000 000, 1 000 000, 500 000, 500 000, 500 000, 500 000, 250 000, 250 000 steps respectively. On all other static environments, we use a regular learning schedule with equal number of steps per skill since they are simple enough and converge in roughly equal speed.

## D.5    BASELINES

**DIAYN**    We created our own implementation PyTorch based on Yarats & Kostrikov (2020), following the official implementation of Eysenbach et al. (2018) method to the best of our ability. We used the same architecture as DISk for actor and critic network. For the discriminator, we used an MLP with two hidden layers of width 256 that was optimized with a cross-entropy loss. To make the comparison fair with our method, the discriminator was given access to the $\sigma(s)$ values in all cases. Each of the DIAYN models were trained with the same number of skills and steps as DISk, as shown in Table 2.

While the performance of DIAYN in complex environments like Ant may seem lackluster, we found that to be in line with the work by Sharma et al. (2019b) (see their Fig. 6, top right.) We believe a

big part of the seeming gap is because of presenting them in a $[-125, 125]^2$ grid. We believe this presentation is fair since we want to showcase the abilities of Off-DADS and DISk.

**Off-DADS** We used the official implementation of Sharma et al. (2020), with their standard set of parameters and architectures. We provide $5\,000\,000$ steps to each environment except otherwise specified, and to evaluate, we sample the same number of skills as DISk from the underlying skill latent distribution. For the hierarchical experiments, we sample the skills and fix them first, and then we create a hierarchical agent with those fixed skills.

### D.6 COMPUTE DETAILS

All of our experiments were run between a local machine with an AMD Threadripper 3990X CPU and two NVIDIA RTX 3080 GPUs running Ubuntu 20.04, and a cluster with Intel Xeon CPUs and NVIDIA RTX 8000 GPUs, on a Ubuntu 18.04 virtual image. Each job was restricted to maximum of one GPU, with often up to three jobs sharing a GPU.

## E CONSISTENCY OF SKILLS

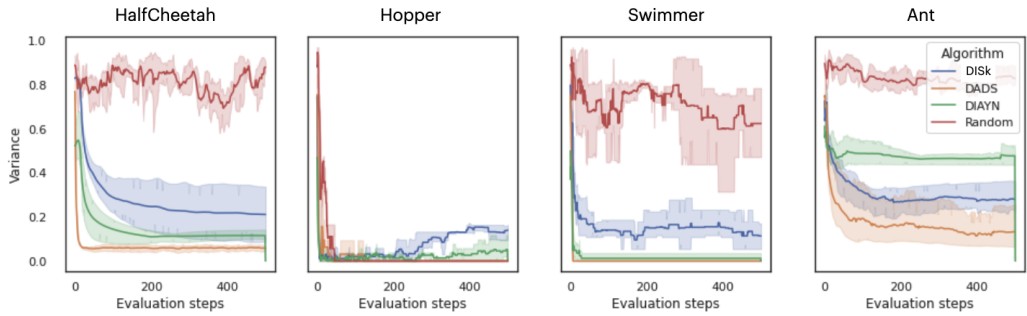

Figure 9: Mean normalized variance over skills of different agents on four different environments (lower is better). On average, the initial position $s_0$ has high normalized variance since the agent has little control over it, and the subsequent steps has lower normalized variance as the skills lead to consistent states.

To ensure that the diversity of the agent as seen from Section 4.2 are not coming from entirely random skills, we run an experiment where we track the average consistency of the skills and compare them to our baseline. To measure consistency, we measure the variance of the agent location normalized by the agent's distance from the origin for DISk, each of our baselines, and a set of random skills as shown in the Figure 6.

We see from this experiment that our method consistently achieves much lower normalized variance than a random skill, thus showing that our skills are consistent. Note that, on the Hopper environment, random skills always crash down onto the ground, thus achieving a very high consistency. While skills from the baselines sometimes achieve higher average consistency compared to DISk, we believe that it might be caused by the higher overall diversity in the skills from DISk.

## F HIERARCHICAL BENCHMARK

Note that in the fast block environment Off-DADS is able to achieve nontrivial performance; this is because as the circle of blocks opens up fast, the agent learns to move slightly left to avoid the last-to-be-opened block to take advantage of the first opening. In the same environment, DISk suffers the most because the environment itself changes too much while a single skill is training, but it is still able to learn a decent set of skills (see Fig. 2.) As a result, their plots seem similar.

## G ABLATION STUDIES

In this section, we discuss the details of the ablation studies discussed on Section 4.5.

## G.1 Reusing the Replay Buffer

Since we use an off-policy algorithm to learn our reinforcement learning policies and compute the rewards given the $(s, a, s')$ tuples on the fly, it is possible for us to reuse the replay buffer from one policy to the next. In this ablation, we examine the efficacy of reusing the replay buffer from previous skills.

Intuitively, if using the replay buffer is effective, it makes more sense that the impact of reusing the replay buffer will be more noticeable when there is a large amount of experience stored in the replay buffer. Thus, for this ablation, we first train six skills on an Ant agent for 4 000 000 steps

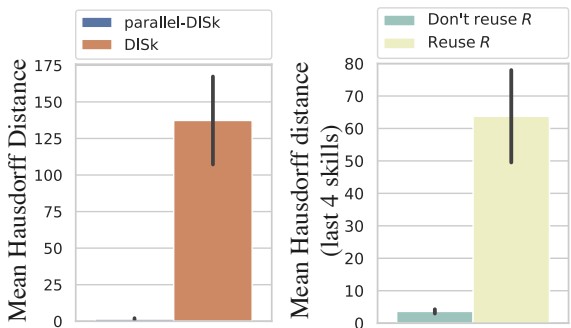

Figure 10: Ablation analysis of DISk on parallel skill discovery and replay $\mathcal{R}$ skill relabelling.

total while reusing the replay buffer, and then train four further skills for 250 000 steps each. In first case, we reuse the replay buffers from the first six skills. On the second case, we do not reuse the replay buffer from the first skills and instead clear the replay buffer every time we initialize a new skill. Once the skills are trained, we measure their diversity with the mean Hausdorff distance metric.

As we have seen on Figure 10, the four skills learned while reusing the buffer are much more diverse than the skills learned without reusing the replay buffer. This result goes to show that the ability to reuse the replay buffer gives DISk a large advantage, and not reusing the replay buffer would make this unsupervised learning method much worse in terms of sample complexity.

## G.2 Parallel Training with Independent Skills

We trained our DISk skills incrementally, but also with independent neural network policies. We run this ablation study to ensure that our performance is not because of the incremental only.

To run this experiment, we initialize 10 independent policies on the Ant environment, and train them in parallel using our incremental objective. We initialize a separate circular buffer for each policy. For each policy, we use the union of the other policies' circular buffer as that policy's $\mathcal{T}$. We train the agent for 5 000 000 total steps, with each policy trained for 500 000 steps. To train the policies in parallel, we train them in round-robin fashion, with one environment step taken per policy per step in the training loop.

As we have seen in Section 4.5, this training method does not work – the policies never learn anything. We hypothesize that this failure to learn happens because the circular dependencies in the learning objective does not give the policies a stable enough objective to learn something useful.

## G.3 Baseline Methods with Independent Skills

To understand the effect of training DISk with independent skills, ideally, we would run an ablation where DISk uses a shared parameter network instead. However, it is nontrivial to do so, since DISk is necessarily incremental and we would need to create a single network with incremental input or output space to properly do so. Still, to understand the effect of using independent skill, we run an ablation where we modify some of our baseline methods to use one network per skill.

Out of our baselines, Sharma et al. (2019a) uses continuous skill latent space, and thus it is not suitable to have an independent network per skill. Moreover, its performance degrades if we use a discrete latent space as shown in Sharma et al. (2019a) Fig. 5. So, we take our other baseline, Eysenbach et al. (2018), which uses discrete skills, and convert it to use one policy per skill.

We see a better performance with Disjoint-DIAYN agent compared to the DIAYN agent, which implies that some of the performance gain on DISk may indeed come from using an independent policy per skill. The reason why disjoint DIAYN performs much better than simple DIAYN may also stem from the fact that it is much harder for the skills in DIAYN to "coordinate" and exploit the

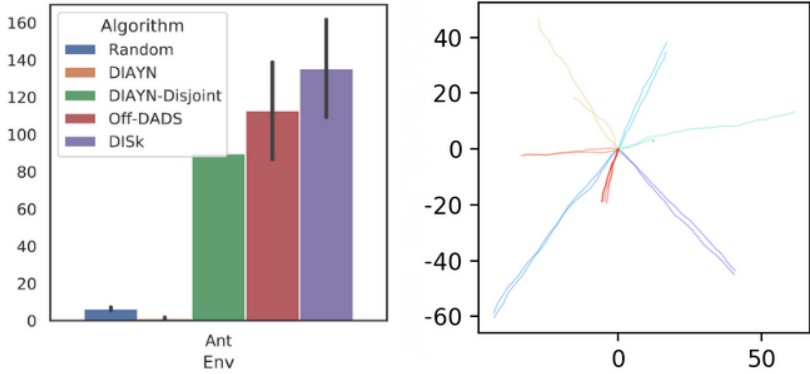

Figure 11: Training DIAYN with one policy per skill improves its performance. On the (left) we show the comparison of mean Hausdorff distance of DIAYN with independent policies alongside our method and other baselines, and on the (right) we show trajectories from running skills from the Disjoint-DIAYN agent.

diversity based reward in a distributed settings where the skills only communicate through the reward function.

### G.4 VALOR-LIKE INCREMENTAL SCHEDULE FOR BASELINES

In VALOR(Achiam et al., 2018), to stabilize training options based on a variational objective, the authors present an curriculum training method where the latents were sampled from a small initial distribution, and this distribution was extended every time the information theoretic objective was saturated. More concretely, if $c$ is a variable signifying the option latent and $\tau$ a trajectory generated by it, then every time their discriminator reached $\mathbb{E}[P_{\text{discrim}}(c \mid \tau)] = p_D$, where $p_D$ is a set constant $\approx 0.86$, they expanded the set of option latents. If there were $K$ latents when the objective saturates, it was updated using the following formula:

$$K := \min\left(\left\lfloor \frac{3K}{2} + 1 \right\rfloor, K_{\max}\right) \tag{8}$$

Where $K_{\max}$ is a hyperparameter that signifies the maximum number of options that can be learned.

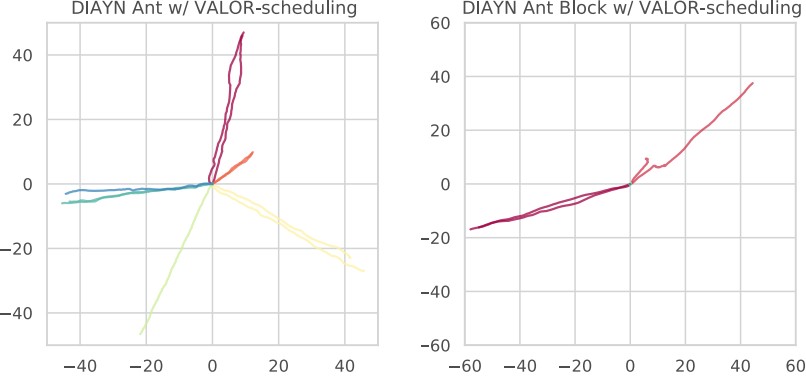

Figure 12: Training DIAYN for 4M steps with curriculum scheduling similar to VALOR(Achiam et al., 2018) results in better performance than vanilla DIAYN but suboptimal performance compared to Fig. 11. On the (left) we show the trajectories from learned skills on Disjoint-DIAYN with an added curriculum training on a static Ant environment. On the (right), we show the same on the dynamic Ant-block environment, where the agent learned two skills that wasn't available at the environment initialization; however all other skills are degenerate.

Adapting this schedule for Off-DADS is difficult since it has nominally infinite number of skills throughout training. However, we implemented this scheduling for DIAYN by sampling the skill

$z$ from the distribution Uniform($K$), starting with $K = 2, K_{\max} = 25$ and updating $K$ with Eq. 8 whenever $\mathbb{E}[P_{\text{discrim}}(c \mid \tau)] \approx p_D$ threshold was reached, where $p_D$ is a hyperparameter. We ran this experiments on top of the disjoint DIAYN (Sec. G.3) baseline and searched over possible $p_D$ values, since we found it necessary to have DIAYN converge to any meaningful skills. We chose the $p_D$ values from the set $\{\frac{n}{n+1} \mid 1 \leq n \leq 10\}$, and found that the best performance was around $6 \leq n \leq 8$, or $p_D \in [0.83, 0.87]$, which lines up with the findings in the VALOR paper.

As we can see on the figures Fig. 12, in complicated static environments like Ant, adding curriculum scheduling to DIAYN performs better than vanilla DIAYN with shared weights, but may actually result in more degenerate skills than Disjoint-DIAYN. We hypothesize this may be caused by the arbitrary nature of the threshold $\mathbb{E}[P_{\text{discrim}}(c \mid \tau)]$, which is also mentioned by Achiam et al. (2018).

On dynamic environments like Ant-block, adding skills successively does allow DIAYN to learn one skill that was not available at the beginning, unlike vanilla DIAYN. However, we believe because of the downsides of curriculum scheduling, it also results in a lot of "dead" skills, which slow down the addition of new skills, which results in few useful skills overall.

# H  DISk Pseudo Code

**Algorithm 2** Pseudocode for DISk training routine for the $M$th skill.

```
# Env: environment
# π: Stochastic policy network
# σ: Projection function
# D: State dimension
# D_σ: Dimension after projection
# M: Current skill number
# P: Number of collected states from each past skills
# max_steps: Number of steps to train the policy for
# replay_buffer: Standard replay buffer for off-policy algorithms
# buf: Circular replay buffer
# α, β: Hyperparameter for normalizing consistency penalty and diversity reward respectively

# Training loop
# At the beginning of training a skill, collect 𝒯 using the learned policies
#   π_m, m ∈ {1, ⋯ , M − 1}
# 𝒯 : P sample states (projected) visited from each of the M-1 previous skills, (M-1)xPxD_σ
𝒯 = []
for m in {1, 2, ..., M−1}:
    total_collection_step = 0
    collected_projected_states = []
    while total_collection_step < P:
        x = Env.reset()
        done = False
        while not done:
            total_collection_step += 1
            a = mode(π_m(x))
            x_next, reward, done = Env.step(a)
            # Add projected state to the collected states
            collected_projected_states.append(σ(x_next))
            x = x_next
    𝒯.append(collected_projected_states)

total_steps = 0
while total_steps < max_steps:
    x = Env.reset()
    done = False
    while not done:
        total_steps += 1
        a ∼ π(x)
        x_next, reward, done = Env.step(a)
        replay_buffer.add(x, a, x_next)
        x = x_next

        # Push projected state to the circular buffer
        buf.push(σ(x_next))

        # Update phase
        for i in range(update_steps):
            # sample a minibatch of B transitions without reward from the replay buffer
            # (x_t, a_t, x_{t+1}): state (BxD), action (Bx|A|), next state (BxD)
            (x_t, a_t, x_{t+1}) = sample(replay_buffer)
            # compute entropy-based reward using the next projected state σ(x_{t+1})
            r_t = compute_rewards(σ(x_{t+1}))
            # train exploration RL agent on an augmented minibatch of B transitions (x_t, a_t, r_t, x_{t+1})
            update_π(x_t, a_t, r_t, x_{t+1}) # standard state-based SAC

# Entropy-based task-agnostic reward computation
# s: state (BxD_σ)
# b: diversity reward batch size
def compute_rewards(s, b = 256):
    𝒯_b = uniform_sample(𝒯, b) # Sample diversity reward candidates

    # Finding k-nearest neighbor for each sample in s (BxD_σ) over the candidates buffer 𝒯_b, (bxD_σ)
    # Find pairwise L2 distances (Bxb) between s and 𝒯_b
    dists = norm(s[:, None, :] − 𝒯_b[None, :, :], dim=-1, p=2)
    topk_dists, _ = topk(dists, k=3, dim=1, largest=False) # compute topk distances (Bx3)
    # Diversity rewards (Bx1) are defined as L2 distances to the k-nearest neighbor from 𝒯_b
    diversity_reward = topk_dists[:, -1:]

    # Finding k-nearest neighbor for each sample in s (BxD_σ) over the circular buffer buf, (|buf|xD_σ)
    # Find pairwise L2 distances (Bx|buf|) between s and buf
    dists = norm(s[:, None, :] − buf[None, :, :], dim=-1, p=2)
    topk_dists, _ = topk(dists, k=3, dim=1, largest=False) # compute topk distances (Bx3)
    # Consistency penalties (Bx1) are defined as L2 distances to the k-nearest neighbor from buf
    consistency_penalty = topk_dists[:, -1:]

    reward = β*diversity_reward − α*consistency_penalty
    return reward
```

# I   DISk Skills Learned In the Static Ant environment In Sequential Order

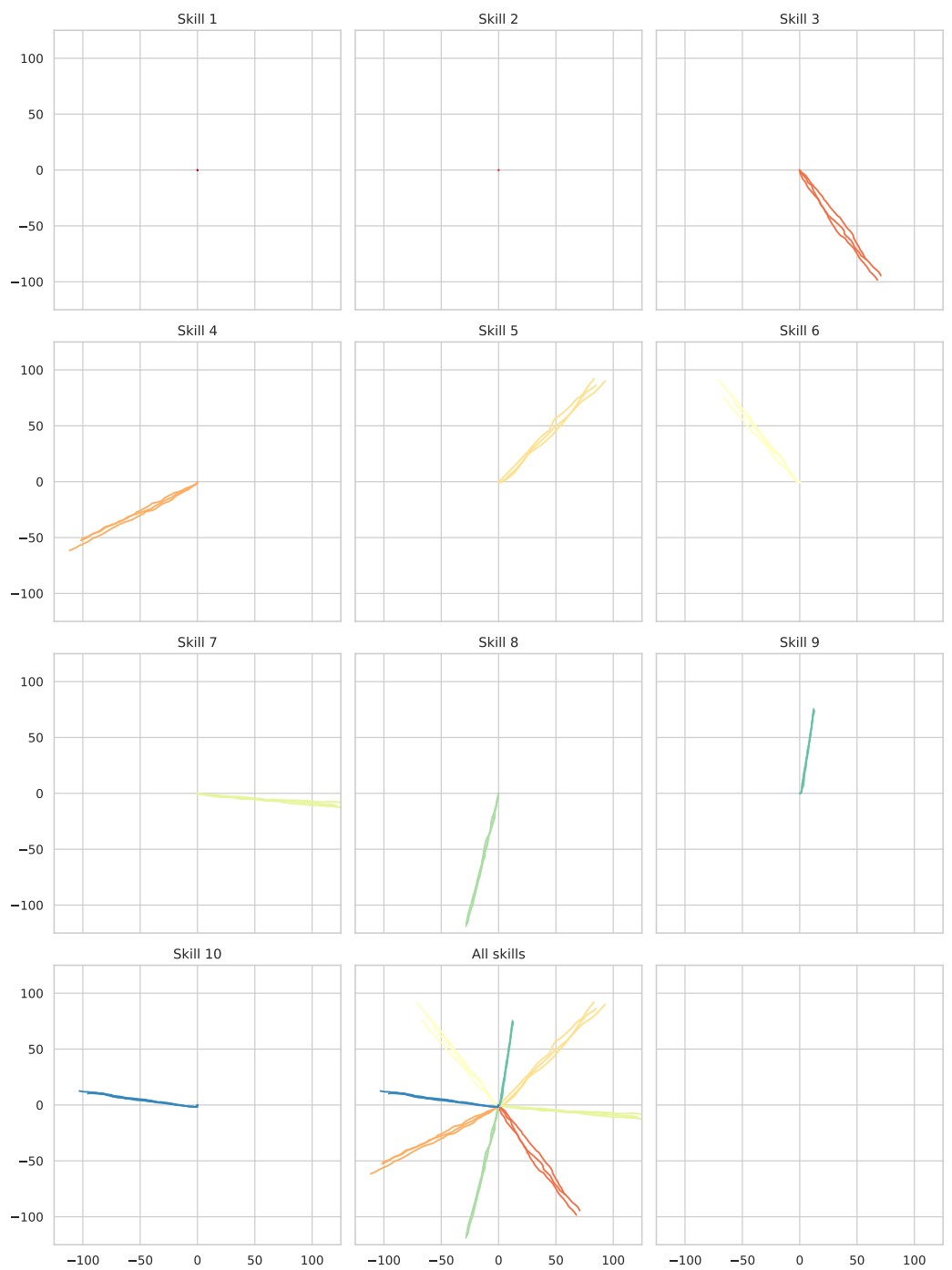

Figure 13: All 10 skills learned by DISk in a static Ant environment in the order learned. Even though it seems like skill 1 and 2 did not learn anything, skill 1 learns to stand perfectly still in the environment while skill 2 learns to flip over and terminate the episode.

