# OpenReview forum: "One After Another: Learning Incremental Skills for a Changing World"
_ICLR.cc/2022/Conference — ICLR 2022 Poster_

### Official Review · Reviewer_bA27 · 2021-10-26

**Correctness:** 4
**Technical Novelty And Significance:** 3
**Empirical Novelty And Significance:** 3
**Recommendation:** 6
**Confidence:** 5

**Main Review:**

**Positives**

- The paper considers a setting that is both novel and relevant. Providing agents with the ability to continuouslly acquire new skills in evolving environments is an important research direction.
- It is well written and easy to follow. The introduction and related work sections properly put the work in context.
- Results are strong and show that DISk can discover more diverse locomotion behaviors than DIAYN and Off-DADS, both under fixed and changing environment dynamics.


**Concerns**

- My main concern is related to the significance of the domains used to evaluate the different methods. I understand that these locomotion environments where agents need to discover skills that walk/run in different directions, so that a hierarchical architecture can then leverage these skills to solve more complex navigation tasks, have been a standard benchmark in the literature for at least 5 or 6 years. However, if we want to truly develop agents that can discover skills autonomously, we need to start considering more complex environments. I cannot agree more with the first sentence of the paper (*Reward-free, unsupervised discovery of skills is an attractive alternative to the bottleneck of hand-designing rewards in environments where task supervision is scarce or expensive.*). Unfortunately, the paper then considers environments where it is very simple to design reward functions that produce the desired behaviors. Note that previous works have considered unsupervised pre-training on Atari (e.g. VISR, APT). Another interesting domain that should be easier to integrate in the current codebase is the Fetch environments in Mujoco, potentially modified to include more objects that could lead to the emergence of manipulation skills.
- It is not surprising that the baselines fail in the presence of changing dynamics (e.g. overfitting to the last final leg). I wonder if authors could introduce slightly stronger baselines. For instance, a version of DADS where the policy is checkpointed before every change to the dynamics. The discovered set of skills would then be the combination of all the checkpoints (e.g. if there are N checkpoints with M skills each, this version of DADS would return $N \times M$ skills). This would show whether the problem with DADS is lack of adaptation or catastrophic forgetting.


**Other comments and questions**

- Please cite [SNN4HRL (ICLR 2017)](https://openreview.net/forum?id=B1oK8aoxe). This is a very relevant work that combines an intrinsic reward proportional to the magnitude of the speed of the robot with a mutual information regularizer that makes skills distinguishable from each other.
- How does DISk behave when the the environment dynamics are changed by placing new obstacles instead of removing them? For instance, by making a wall similar to the one in Figure 1 [here](https://arxiv.org/pdf/1712.06560.pdf) spawn after a certain number of iterations?
- Figure 9 (right) shows the MHD after a certain number of steps. It would be more insightful to see a plot showing how the MHD evolves as a function of the number of steps.
- I was surprised to see that Skill 2 in Figure 10 does not move at all. Could authors please provide their intuition for why this happens?
- Please provide a reference for the following sentence in the introduction:
> Not only that, but the agents also generalize poorly to any changes in the environment


**Summary Of The Paper:**

The paper introduces DISk, an information-theoretic skill discovery method that discovers skills in a sequential fashion instead of all at once. This provides better adaptation to changes in environment dynamics. Evaluation is performed on standard continuous control benchmarks under changing and constant dynamics, and DISk compares favorably to baselines like DIAYN and Off-DADS.

**Summary Of The Review:**

Overall, I lean towards accepting the paper but I have serious concerns about the adequacy of the evaluation protocol. While they clearly prove the advantages of DISk, I'm not convinced that by making advances in these toy environments we are truly getting closer to developing agents that can acquire useful skills autonomously. I acknowledge that the paper mostly follows standard practice in the community, which is why I still recommend acceptance, but the submission would be much stronger if authors could provide results in other settings (e.g. Fetch, Atari).

---

> ### Author Response · Authors · 2021-11-12
> **Response to the Official Review of Paper1845 by Reviewer bA27**
>
> First of all, we would like to thank you for taking the time to review our work, placing it in the context of the current state of the field. We are also glad that you find our research direction important, and our work strong yet easy to follow. Below, we address the questions you have raised below in detail.
>
> **Significance of the evaluation domains:** We agree with you in the sense that the skill learning methods are yet to be battle-tested in applications where it's hard to hand-design goals. However, we are excited about DISk since it takes something as simple as a distance metric between two observations, and converts it to a set of incremental skills which can even be learned in a changing environment. We can easily imagine how, paired with an appropriate representation learning module, methods like DISk can be applied to even image based learning. However, that requires us to first establish the method upon which the community can build, which requires us to show the effectiveness of such methods in relatively straightforward tasks first.
>
> **Failure from lack of adaptation vs. catastrophic forgetting**: While the additional baseline you propose with checkpointed Off-DADS may indeed create a stronger baseline, we believe that the experiments we present in the paper, and some additional experiment that we ran at your request, show that baselines like Off-DADS suffer from _both_ lack of adaptation AND catastrophic forgetting.
>
> In the Block experiment, as there is only a single gap in the block-circle in the beginning, the DADS environment model overfits to that single gap, and the agent only learns to move in that single direction. Later on, even when more of the block circle opens up, the DADS agent does not learn to explore other directions since the model error is very low and the intrinsic reward is already very high. Thus, it fails to discover more diverse skills as it lacks adaptability.
>
> However, when we force the DADS agent to adapt by breaking a leg of the ant agent (and thereby driving the model prediction error high), DADS first adapts, and then overfits all of its skills to the newly broken leg. Concretely, the skills from DADS broken-leg agent 0.3 M steps after introducing the latest broken leg (so 9.3M step total) looks like the one on [left figure here](https://i.imgur.com/o1xTiJL.png). Compare this figure with Figure 3 in the paper (reproduced on the right) showing the final skills at the end of training. This comparison clearly shows that Off-DADS agent has adapted first, and then catastrophically forgot the earlier learned skills for other broken legs. We have added a figure to the Appendix (Fig. 7 and 8 in the revised paper) that shows the same.
>
> Combining these two experiments show us that DADS can potentially have a difficult time adapting to new environments or remembering old ones based on the way the environment changes.
>
> ### Response to other comments:
> * **Citation for SNN4HRL**: Thank you for the pointer, we have added the citation.
> * **Added obstacles case**: We believe the behavior should be similar to the broken leg case, since both of them are changes in the environment/agent that makes some previously feasible movements infeasible. We are currently running this experiment and will add it to the paper when completed.
> * **MHD metric over time**: We have found that in higher (> 1) dimensions, comparing the MHD metric between two sets with different numbers of clusters is not as reliable, which is why we only show the MHD at the end of training with the same number of skills.
> * **Skill 2 behavior**: Due to how we define our diversity reward as 1 where there are no previous skills, our first skill is encouraged to stay alive for as long as possible, while the second skill has no such motivation except to be different from the first skill. Thus, the first skill in Figure 10 learns to stand perfectly still while the second skill learns to just flip over and end the episode.
> * **Reference for a statement in the introduction**: We have addressed this in the updated paper; thank you for pointing out our oversight.
>
> In the light of the above clarifications, we would like to ask if you are willing to increase your score assuming we have addressed your concerns. Otherwise, please let us know if you have additional questions.

---

> > ### Comment · Reviewer_bA27 · 2021-11-18
> > **Discussion**
> >
> > I had not replied before because there are some experiments running and I was (and still am) waiting to have all the information before making a final decision. I should have probably replied earlier to let authors know, apologies for that. I would like to thank the authors for replying to my concerns thoroughly and for reporting new results that I found quite insightful: it's interesting to see that DADS can suffer from both lack of adaptation and catastrophic forgetting, as well as how much DIAYN improves when not sharing weights across skills.
> >
> > My main concern is still related to the significance of the evaluation domains; while I agree with authors that ideas need to be evaluated in relatively straightforward domains first, I would also like to note that the unsupervised RL community is (slowly) turning towards more complex domains. It is not uncommon to see this type of methods evaluated on Atari (e.g. VISR, APT, [EDDICT](https://proceedings.neurips.cc/paper/2021/hash/5f7f02b7e4ade23430f345f954c938c1-Abstract.html)), a domain which would be much more challenging for DiSK because it would need to train a separate CNN for each skill without any type of parameter sharing. Note that I am not saying that there aren't ways to overcome this problem, and indeed I share the authors' intuitions, but the version of DiSK presented in this submission would suffer from this issue.
> >
> > While I'm not convinced about increasing the score to a strong accept for the aforementioned reasons, I still lean towards acceptance -- more so than before the rebuttal due to the new results.

---

> > > ### Author Response · Authors · 2021-11-29
> > > **Possible extensions on different domains**
> > >
> > > We appreciate your feedback, and acknowledge the fact that the domains like ATARI, beyond gym locomotions, present a unique challenge to skill learning algorithms. We agree that the added challenge of learning a visual encoder would create an interesting setting. For a better setting to understand skill learning in a changing environment, we believe the different procedurally generated levels in the ProcGen environments might be more appropriate, interesting, and relevant to real progress than the static environments of ATARI. However, given the nontrivial cost of computation for running such experiments, we believe it to be more appropriate for future work.

---

> > > > ### Comment · Reviewer_bA27 · 2021-11-30
> > > > **Re: Possible extensions on different domains**
> > > >
> > > > Absolutely. I only mentioned Atari because it has been used in related works. I am not aware of any paper evaluating skill discovery methods on ProcGen, but I totally agree that it would be a great evaluation benchmark in which DiSK should work well as long as a good solution is found for overcoming the representation learning issues. Please note that it was not my intention to ask authors for such results during the rebuttal process; I was just flagging what I consider one of the weakest points of the paper. This said, I'm still recommending the paper for acceptance.

---

### Official Review · Reviewer_HdqY · 2021-10-31

**Correctness:** 4
**Technical Novelty And Significance:** 3
**Empirical Novelty And Significance:** 3
**Recommendation:** 6
**Confidence:** 3

**Main Review:**

### Pros:
1. The paper addresses an important problem in RL: unsupervised skill discovery in stochastic or evolving environments. Dropping this simplifying assumption brings it closer to real-world environments. As far as I'm concerned, this is the first work in this space. To me, the problem itself is real and practical.
2. The proposed framework of incremental skill discovery (DISk) is novel for separating the learning of individual skills.
3. This paper provides comprehensive experiments, including both qualitative analysis and quantitative results, to show the effectiveness of the proposed framework.

### Cons:

1. One thing I think was missing is the time-wise comparison between the results of DISk vs baselines in the main body of text. How long did the training take for DISk, DIAYN and Off-DADS in terms of both environments steps and wallclock time on the results presented in Section 4.2 - 4.4? DISk needs to train many neural networks (instead of one like in baselines). How slow does this make them compared to other methods (i.e., in terms of time when trained on the same number of environment interactions)? This is especially relevant for Section 4.4: transferring skills to the downstream task. I hope the authors can elaborate on this.
2. Also, I'm not particularly convinced by the ablations performed in the paper. The authors try to examine whether "the performance gains in the static environment are primarily due to the independent policy part", isn't the natural test for this training shared policy in parallel, rather than individual ones?
3. Have the authors considered running a version of baselines where the network weights aren't shared among the skills? Authors themselves wonder whether "the performance gains in the static environment are primarily due to the independent policy part", after all. This additional experiment might shed light on the question.

Some typos:
- Page 4: paragraph 3: "used" → "using"

## Post-rebuttal discusion

I thank the authors for the detailed response. Based on the clarifications that the authors provided, I've decided to increase my score by 1.

**Summary Of The Paper:**

The authors have presented DISk, an unsupervised skill discovery method that incrementally learns a sequence of skills. This allows DISK to adapt to changes in the environment or agent dynamics during training. The authors demonstrate experimentally that in both evolving as well as static environments, DISk outperforms existing skill discovery methods on both skill quality and the ability to solve downstream tasks.

**Summary Of The Review:**

To me, the problem is real and practical. The methods that the authors propose is novel. Authors provide comprehensive experiments to supports the proposed method's superiority compared to existing baselines. Nonetheless, I have several concerns regarding fair comparison of the DISk vs existing methods. Specifically, I'd learn more about its computational efficiency and see additional experiments and ablations.

---

> ### Author Response · Authors · 2021-11-12
> **Response to the Official Review of Paper1845 by Reviewer HdqY**
>
> We would like to thank you for your thoughtful comments. It seems that despite finding our work technically and empirically novel, you have some reservations about our work. These are important questions, and so we discuss them in detail below and have also updated the text of the main paper.
>
> **Computational Efficiency:** On all environments in all our experiments, DISk and all our baselines are provided with exactly the same number of environment interactions -- that is how we compare their behavior in a standardized way. Informally, on average for every 1M environment interaction on the Ant environment, DISk took us about 2.5 hours, Off-DADS took 4.2 hours, and DIAYN took 3.3 hours, in terms of wall clock training time. Even though DISk is initializing a new network for every skill, it is training them in a model free way, while Off-DADS needs to train a forward dynamics model and DIAYN needs to train a discriminator. However, we did not present this in the paper since we used the official Off-DADS implementation written in Tensorflow while our DISk and DIAYN implementations are in Pytorch.
>
> **Parallel, shared-parameter DISk:** Please see our [general comment here](https://openreview.net/forum?id=dg79moSRqIo&noteId=ocHjRoPnGb0) where we address this concern. To summarize, since DISk uses independent policies trained sequentially, training shared policies parallely would require two changes in a single ablation. Even if training shared policies in parallel with the DISk intrinsic reward were to give us better performance in static environments than its current incarnation, it would lose its incremental nature, which would go against our entire continual learning premise.
>
> **Baseline without sharing weights:** Please see our general comment here [general comment here](https://openreview.net/forum?id=dg79moSRqIo&noteId=ocHjRoPnGb0) where we address this concern. To summarize, we chose independent networks primarily because we needed to create an agent with a set of policies that are learned incrementally. The major baseline that we compare against is Off-DADS, which in its standard version is conditioned by a real number interval and thus fundamentally can’t accommodate independent policies for its nominally infinite number of skills. If we were to modify it first by giving it discrete skill space, it would create a significantly inferior variant of the algorithm as shown by the original DADS paper. Per reviewer comments, we have added an ablation with DIAYN with independent weights, which performs better than the original DIAYN but still performs worse than Off-DADS or DISk.
>
> We hope our comments above have assuaged your concerns -- we look forward to hearing your follow-up thoughts and hope you can come to share our views about the contributions of DISk to the field. Finally, in the light of the above clarifications, assuming we have addressed your concerns, we would like to ask if you would be willing to increase your score. Otherwise, please let us know if you have additional questions -- we are happy to give any further clarification.

---

> > ### Comment · Reviewer_HdqY · 2021-11-26
> > **Responce the the rebuttal**
> >
> > I thank the authors for the detailed response. Based on the clarifications that the authors provided, I've decided to increase my score by 1.

---

> > > ### Author Response · Authors · 2021-11-29
> > > **Thank you for the response**
> > >
> > > We thank you for increasing your rating to an Accept. We welcome any further questions or comments that you may have that you believe will strengthen the work while the discussion period is open.

---

### Official Review · Reviewer_eZE6 · 2021-11-02

**Correctness:** 3
**Technical Novelty And Significance:** 3
**Empirical Novelty And Significance:** 3
**Recommendation:** 6
**Confidence:** 4

**Main Review:**

STRENGTHS

The authors present three features that each might be useful for the unsupervised skill learning literature. Their skill visualizations are well-presented.

WEAKNESSES

The current version of this paper suffers from unclear motivation, missing baselines, and missing ablations. I do believe there are good ideas here, however the paper is not ready for publication.

**Unclear motivation**: The authors motivate their approach by describing skill learning in a slowly evolving environment, where it is important to remember skills for every version of the environment along the way. This strikes me as a quite artificial setting. If the world has evolved, why wouldn't we care about the agent's performance just in the most recent "current" world? If instead the focus is indeed on performing well in many different environments, why not randomly sample one each episode? Depending on the answer to this question, it would e.g. be interesting to rerun the experiment in Figure 3 but with the broken leg resampled every episode, rather than every 1M steps.

Incremental skill learning is a good idea; so good in fact that it's been done before.

**Missing baseline #1: VALOR**: Achiam et al 2018 (https://arxiv.org/abs/1807.10299) introduced a curriculum for learning skills which introduced new skills based on discriminability performance on current skills, rather than at preset intervals. It is an important baseline as previous work, but also to allow the authors to compare performance-based and fixed-interval skill expansion.

**Missing baseline #2: DSP**: Zahavy et al 2021 (https://arxiv.org/abs/2106.00669) introduced Diverse Successive Policies, which incrementally learns new skills that maximize diversity in terms of successor features. This alternative take on fixed-interval skill expansion is also an important baseline.

The approach the authors propose simultaneously introduces three features to unsupervised skill learning, but the current experiments don't tease apart which matter. Ideally, I would like to see *each* combination of these features tried with each baseline, however that be a lot to ask and for example how exactly to adapt DIAYN to learn skills incrementally involves a lot of non-trivial choices. However, one of these features is very easy to try with each baseline.

**Missing baseline #3: all other baselines with one network per skill**: It is an unfair advantage to DISk to get to use a separate network per skill, while other algorithms do not. Each other baseline should be run as two versions - one with a shared network, and one with a network per skill. As the authors rightly identify, it may be that one or the other approach is preferred in specific environments. Additionally, the authors should emphasize more clearly that despite repeatedly using the motivation of life-long learning, an algorithm whose parameter use grows linearly in time seems problematic for that setting.

I would also like to clarify one existing baseline and one feature of DISk.

**Question: why is DIAYN so bad?** DIAYN seems to perform implausibly bad, especially in the Ant environments. For example, if DIAYN is performed on velocity-based features, wouldn't the policies in Figure 2 move? My suspicion is that the authors use velocity-based features *in addition* to the environment-provided features, rather than *instead of* them. If true, this is now an unfair comparison, since of course the DISk skills that are *only* based on velocity diversity will learn to move, while the DIAYN skills can instead manifest as different joint configurations. Thus DIAYN should be rerun on velocity-based features alone. If my suspicious is wrong, however, hopefully the authors can provide an alternative intuition for what is going on.

**Question: how is the skill expansion interval chosen for DISk?** It seems like the interval at which new skills were introduced was hand-crafted to match the interval at which the environment evolves, or in the static environment case, was carefully tuned to a complicated variable-interval schedule (Appendix D.4) to get interesting skills. This again seems quite artificial and unfair to the baselines. How sensitive is DISk to this schedule? Seems like an important ablation. Additionally, how do the authors imagine DISk being used in practice in environments where the user doesn't control the environment dynamics?

In addition to the ablation suggested in the previous question, there is another one missing.

**Missing ablation: shared network**: in section 4.5, the authors rightly state "Compared to prior work, DISk not only learns skills incrementally, but also learns independent neural network policies for each skill." However, the authors then only ablate the first feature, and not the second. The authors should test a version of DISk that uses a single shared network of similar architecture to the baselines.

Additional more minor comments and suggestions:
1. Another interesting approach to compare to is that of Groth et al 2021 (https://arxiv.org/abs/2109.08603), which uses a curiosity bonus alone to drive exploration and then periodically freezes copies of the policy as skills. This approach shares with DISk a gradually evolving continuous development of skills that is targeted to the life-long learning setting. The work seems to have been published after the ICLR deadline, however, so it would be unfair to ask for it as a baseline here. That said, it is obviously a baseline of interest for future versions of this work, and would make for good discussion in the present related work section.
2. The notation of section 3.2 seems to confuse random variables with the probability distributions they are drawn from. Additionally, the union operation in equation 6 doesn't seem well-defined to me. I think what the authors actually mean is: S_m ~ p(s|pi_m), S~(1/M)sum_m p(s|pi_m), and the first entropy term in equation 6 should then be simply H(S).
3. The authors motivate two of DISk's features (incremental skill learning, independent skill policies), but not the third - why do the authors use a seemingly complicated new entropy estimator, rather than for example using the variational (discriminator-based) approach of DIAYN (i.e. breaking down I(S;Z) in the other direction)?
4. Why isn't DIAYN included in Figure 3?

**Summary Of The Paper:**

The authors introduce a new unsupervised skill learning algorithm that combines three features: 1) incremental skill learning (i.e. learns one skill after another), 2) independent networks per skill, and 3) a nearest neighbor entropy estimator. They compare their approach to DIAYN and off-DADS in both static and dynamic MuJoCo environments in OpenAI gym. Comparisons involve skill visualizations, state coverage metrics, and reusing the skills for HRL on a goal-seeking task.

**Summary Of The Review:**

Adapting skill learning approach to dynamic worlds and learning skills incrementally are good ideas, however the current set of experiments do not sufficiently distinguish the author's approach from existing literature, nor pinpoint what is important about their setup. With the baselines, ablations, and clarifications above included, I think this will be a strong paper that I look forward to rereading.

---

> ### Author Response · Authors · 2021-11-12
> **Response to Official Review of Paper1845 by Reviewer eZE6 [2/2]**
>
> **Performance of DIAYN on Ant:** DIAYN on Ant _was_ indeed trained on velocity features alone, which is both the sensible approach and what you said it should be. As for the less-than-stellar performance in complex environments, we believe it is because instantaneous velocity is both easily controllable (although noisily and in small magnitudes) and easily discriminable, especially when there are only 10 skills to discriminate and a shared network to coordinate them all. As you have correctly deduced, DIAYN is very much dependent on the discriminator performance. When the discriminator achieves perfect, zero error discriminability when the skills have learned to crash in 10 different directions, they have no reason to learn any further. We show in the new ablation study that we can get around this by splitting it over independent networks, and thus preventing the skills to “cooperate” in any way except through the discriminator. In our experiments, any changes to the DIAYN behavior while keeping the original algorithm requires us to modify the environment further in DIAYN’s favor, or provide the agent location or noisy estimates of velocity to the discriminator, which creates an unfair comparison vs. Off-DADS and DISk. Finally, DIAYN wasn’t included in Figure 3 since we only compared DISk with Off-DADS in the changing environments as those are the only algorithms that performed well in static environments.
>
> **Skill expansion interval:** We must clarify an important confusion. You asked:
> > Additionally, how do the authors imagine DISk being used in practice in environments where the user doesn't control the environment dynamics?
>
> We believe there must have been a misunderstanding of our experimental setup. In the Block experiment, we operate in the setting where DISk has no a priori knowledge of the environment changing, it can only observe those changes through interactions. In the broken leg experiment, we introduce a new skill with the agent’s physical dynamics change, which can also be automated without a-priori knowledge or control over the environment.
>
> The skill expansion intervals presented in Appendix D4. is less of a hand-tuned schedule and more of a guide for sake of replicability of how long it took the skills (as visualized in Appendix Fig. 11) to converge. As you can see in Fig. 11, the first two skills learn trivial behavior and thus take a short time to converge, and after that complex behaviors take an ever decreasing time to learn. We have updated the Appendix to clarify this concern. Moreover, in the Block experiments, we show the robustness of DISk to the learning schedule by comparing an agent with the same schedule in three different environments with different rates and magnitudes of change. Furthermore we have ablated in Fig. 2 (middle) and shown how much faster changes in the environment leads to degradation in DISk performance (which however is expected from standard continual learning algorithms, and to some certain degree, humans.)
>
> In summary, we hope that our clarifications help address some of the confusion around our work. We have made every effort to modify the text in the paper to make these points clear. In light of our clarifications above, we would like to politely ask if you are willing to increase your score assuming we have addressed your concerns. Otherwise, please let us know if you have additional questions.
>
> [1] Li, Z., & Hoiem, D. (2017). Learning without forgetting. IEEE transactions on pattern analysis and machine intelligence, 40(12), 2935-2947.
> [2] Ruvolo, P., & Eaton, E. (2013). ELLA: An Efficient Lifelong Learning Algorithm. ICML.
> [3] Ammar, H. B., Eaton, E., Ruvolo, P., & Taylor, M. E. (2015, February). Unsupervised cross-domain transfer in policy gradient reinforcement learning via manifold alignment. In Twenty-Ninth AAAI Conference on Artificial Intelligence.
> [4] Tanaka, F., & Yamamura, M. (1997, August). An approach to lifelong reinforcement learning through multiple environments. In 6th European Workshop on Learning Robots (pp. 93-99).
> [5] Wilson, A., Fern, A., Ray, S., & Tadepalli, P. (2007, June). Multi-task reinforcement learning: a hierarchical bayesian approach. In Proceedings of the 24th international conference on Machine learning (pp. 1015-1022).
> [6] Bendale, A., & Boult, T. E. (2016). Towards open set deep networks. In Proceedings of the IEEE conference on computer vision and pattern recognition (pp. 1563-1572).
> [7] Fei, G., Wang, S., & Liu, B. (2016, August). Learning cumulatively to become more knowledgeable. In Proceedings of the 22nd ACM SIGKDD International Conference on Knowledge Discovery and Data Mining (pp. 1565-1574).
> [8] Fernando, Chrisantha, et al. "Pathnet: Evolution channels gradient descent in super neural networks." arXiv preprint arXiv:1701.08734 (2017).
> [9] Chen, Z., & Liu, B. (2018). Lifelong machine learning. Synthesis Lectures on Artificial Intelligence and Machine Learning, 12(3), 1-207.

---

> > ### Comment · Reviewer_eZE6 · 2021-11-22
> > **Thank you for your response**
> >
> > Thank you for your thorough response. I found it helpful, as well as the responses to other reviewers.
> >
> > Thank you for clarifying the problem setting you have in mind, as well as how that affects your algorithmic choices. I have more confidence in both the relevance of the problem you are tackling, as well as the motivation behind your model choices (e.g. the entropy estimator).
> >
> > Thank you also for clarifying the details of the DIAYN setup and for providing intuition on why it fails. The new one-network-per-skill version of DIAYN is also a great addition to the paper, and is important enough that I recommend moving it to the main text.
> >
> > Thanks for adding the references, although please note the VALOR reference seems to incorrectly cite Florensa et al 2017 (https://arxiv.org/abs/1704.03012) rather than Achiam et al 2018 (https://arxiv.org/abs/1807.10299). On further reflection, I agree with your point that DSP is a different enough setting (i.e. they jointly maximize task reward and diversity) so as to not be an important baseline for you, however I disagree on not including VALOR, and think your paper would much be stronger for it. While VALOR doesn't seem to be peer-reviewed, it is a well-known paper with 78 citations (according to Google Scholar) and in my opinion is a foundational piece of work in the variational infomax-style skill learning literature. While it is true that VALOR cannot in principle introduce an arbitrary number of skills like DISk, in practice one can set a high maximum number of skills for the scale of experiments the field is currently dealing with. The point of including this baseline is that it still uses the discriminator-style skill learning of DIAYN, but also introduces skills incrementally, thus perhaps providing a stronger challenger from that literature. While the original implementation may not have been open sourced, the specific component I'm suggesting using (the expanding skill curriculum of equation 5) is easy to implement alongside your DIAYN implementation (I've done it myself in multiple codebases). To clarify, I don't think it is necessary to reproduce the VALOR method in full; the expanding skill curriculum is the interesting point of comparison for your paper, and can be e.g. used with DIAYN.
> >
> > On the skill expansion interval, I understand that the DISk agent itself doesn't have access to the environment change frequency - what I meant is that you the experimenter do, and this knowledge seemed to be factored in to the scheduling. If what you tried to do was to select the intervals to match skill convergence, why not just use a performance-based skill expansion trigger like in the VALOR paper?
> >
> > Considering your responses and the additions to the paper, I'm raising my score to a weak reject for now. My score would be at least a weak accept with a VALOR baseline included. My score would also be higher if the evaluation environments were more interesting (per the comments of Reviewer bA27).

---

> > > ### Author Response · Authors · 2021-11-22
> > > **Additional experiments with VALOR**
> > >
> > > We thank you for taking the time to consider our rebuttal and engaging in thoughtful and productive discussion -- we believe your feedback has made “One after Another” a better paper. Please find an update and a response to your further comments below, which we hope you will find satisfactory.
> > >
> > > **VALOR-like expanding skill scheduling/curriculum:**  Firstly, we apologize for the mistaken citation and we have corrected it now. We have added the expanding curriculum learning to DIAYN and added it as another baseline in Appendix G4. As you have correctly mentioned, independent weights and curriculum scheduling improves upon vanilla DIAYN. Unfortunately, it still performs worse compared to DISk. We ran this experiment on both the static Ant environment and the Ant-block setup. In a static Ant environment, we see [the skills presented here (left)](https://imgur.com/a/Q5Zm50s), while in Ant-block we see [these skills (right)](https://imgur.com/a/Q5Zm50s). Unfortunately, we found that with the default parameters from VALOR Sec 3.3, expanding DIAYN performs worse than DIAYN with disjoint weights and a fixed skill set on a static environment, with expanding DIAYN learning a lot of degenerate skills. We also tried it in a dynamic environment, namely Ant-block. There, too, the number of “dead” skills puts to question the usefulness of the curricula in its current form.
> > > It seems like the frequent domain shifts for the discriminator and lack of weight sharing in actor policies slow down and hamper the performance of DIAYN compared to learning all skills simultaneously.
> > >
> > > **VALOR-like trigger for adding skills:** The above experience also leads to a natural response to your question about our scheduling: we found the fixed probability threshold trigger of VALOR to be somewhat unreliable for DIAYN -- threshold too low means new skills are added too fast, threshold too high means new skills are never added, and the right value of the threshold also seems to depend on the current number of skills. While a more principled variant of the same could also work for DISk, we are as of yet uncertain of what that may look like.
> > >
> > > **Disjoint-DIAYN baseline:** We thank you for the suggestion for adding the Disjoint-DIAYN variant on the main paper. We are currently running additional seeds on this experiment and will add it to the main paper when ready.
> > >
> > > We thank you once again for the extremely valuable feedback you have provided us in this rebuttal period, and hope that in light of the updates above you will also update your score of our submission accordingly.

---

> > > > ### Comment · Reviewer_eZE6 · 2021-11-23
> > > > **New VALOR experiments**
> > > >
> > > > Thank you again for the new experiments and discussion! I agree that they improve the paper, and will correspondingly raise my score to a "weak accept." I do this in good faith assuming that the new experiments will be integrated into the main text when they are finished, and that a bit of additional effort will be put into tuning the VALOR baseline. In particular, I think it is worth doing a small sweep over the skill expansion threshold. While I agree that the place to start is with the published value from the original paper, it may be a quite environment-sensitive parameter.
> > > >
> > > > I agree that coming up with more principled skill expansion triggers for DISk is an interesting direction. I also agree that a VALOR-like discriminator-performance-based trigger is not the final answer, though I still think it could be an interesting variant to try, if only to illustrate to your reader where and why it succeeds/fails relative to a fixed-period trigger.

---

> > > > > ### Author Response · Authors · 2021-11-29
> > > > > **Thank you, and following up**
> > > > >
> > > > > We thank you for your good faith. We have already started further runs of the experiments on the disjoint DIAYN baseline and the VALOR-scheduling baseline with a range of skill expansion thresholds. What we have found since the update deadline matches our expectation since the latest update of our paper, and we look forward to updating the camera-ready version with the full results.

---

> ### Author Response · Authors · 2021-11-12
> **Response to Official Review of Paper1845 by Reviewer eZE6 [1/2]**
>
> We would like to thank you for writing such a long and detailed review. We believe there has been a gross mischaracterization of our work, which we will address below. Further, we have updated the paper to reflect this discussion, including a new ablation experiment, additional references, and your suggestion about mathematical notations.
>
> **Unsupervised skill learning with three features:** Inherently, DISk’s first and foremost aim is to discover skills despite changes in the environment. This aim necessitates using incremental skill learning. The fact that we use independent networks per skill and a non-parametric mutual information estimators are both indispensable design choices we had to make to adapt prior works into the continual setting -- neither a shared network nor a parametric entropy estimator (discriminator for DIAYN or forward dynamics model for DADS) can work with a growing number of skills. The fact that our algorithm may work well in a static setting is relevant so far as the set of static environments is a subset of the set of changing ones. We hope that this insight about our focus clarifies the confusion about what DISk is trying to solve.
>
> **Standards in Continual/Lifelong learning:** It seems that some of our design decisions and experimental set-ups that you do not agree with are simply standard practices in continual/lifelong learning literature. For example:
> >  The authors motivate their approach by describing skill learning in a slowly evolving environment, where it is important to remember skills for every version of the environment along the way. This strikes me as a quite artificial setting.
>
> This is the standard setting followed by [1-3, 6-7], and many more examples that you can find in [9]. As RL practitioners, we are inspired by the real world setting where your locomotion robot’s sensors may slowly drift out of calibration, or limbs of your agents may break and get fixed over a long horizon in our broken leg experiment. Our Blocks environment is inspired by incremental expansions in the environment, where for example more rooms of a house can be made available to a domestic robot.
> Similarly, you raise a concern:
> > Additionally, the authors should emphasize more clearly that despite repeatedly using the motivation of life-long learning, an algorithm whose parameter use grows linearly in time seems problematic for that setting.
>
> Which is in fact one of the common design choices in continual/lifelong learning, as seen in [2-5, 7]. We have updated the Background and Related Work sections to reflect this topic. Even with the additional parameters, DISk takes about 2.5 hours per 1M environment interactions, as opposed to 3.3 for DIAYN and 4.2 for DADS, since it does not need to train any additional networks (discriminator for DIAYN and forward dynamics model for DADS), which may also be relevant in a lifelong setting.
>
> **Missing baselines and ablations:** Thank you for pointing out these works. We have added references to them in our paper. However, we would like to point out that neither of the papers you have mentioned have been peer-reviewed, and more importantly, neither have released a codebase to compare against. VALOR uses a curriculum of skills to reduce training instability, which we believe is partially caused because of using a recurrent network for entropy estimation. But we agree that this unintended design choice may make it more suited to adaptation compared to DIAYN. As for Diverse Successive Policies, we are unsure how this is relevant beyond being incremental, since it focuses on learning in a static environment with a provided goal function. Finally, we have addressed your comment about the one network per skill in the [general comment here](https://openreview.net/forum?id=dg79moSRqIo&noteId=ocHjRoPnGb0), but to summarize: our main comparison, Off-DADS uses continuous skills, and thus it is not trivial to simply extend it to one network per skill. We have added an ablation (Sec. G3) showing DIAYN with independent networks, which improves over original DIAYN but not beyond Off-DADS or DISk. Similarly, using a shared network is impossible where DISk needs to learn an incrementally large number of skills, which may be a reasonable ablation for skill learning in a static environment but not in a dynamic one.

---

### Official Review · Reviewer_N9w1 · 2021-11-02

**Correctness:** 3
**Technical Novelty And Significance:** 3
**Empirical Novelty And Significance:** 2
**Recommendation:** 6
**Confidence:** 4

**Main Review:**

## Strengths

- The paper combines two interesting and relevant problems: unsupervised learning of skills and learning in continually changing environments.

- The writing of the paper is clear and the method easy to understand. While a few details are missing (see questions below), the overall approach is simple and can be easily grasped from the explanations. It is also put well into context of prior works.

- The figures visualize the diverse learned skills in an intuitive and easily understandable way.



## Weaknesses

### Conceptual Weaknesses
In comparison to prior works in unsupervised skill discovery the idea of incrementally learning skills with separate networks has a few drawbacks:

- First off, the problem and goal of learning skills in continually changing environment does not seem clearly defined to me: what do we want to final set of learned skills to look like? Should they all work on the final training environment? Should they all work on all environments seen during training? Or should some work on a subset of environments while others work on another subset? What environment would we evaluate the downstream learning in? The paper does not clearly articulate these goals.

-  The requirement to learn a full new network per skill, while feasible in the simple tested environments, seems not scalable to large skill sets. At the same time it seems wasteful to train each new skill from scratch. Some form of parameter sharing between the skills seems required but the very concept of *not* sharing parameters seems to be at the heart of the proposed method and thus in conflict with these considerations.

- The iterative nature of the proposed approach requires us to define a schedule at which we start training new skills. This schedule seems to be fundamentally environment dependent and linked to the rate of change in the environment and thus requires additional human design effort which prior works that learn skills jointly do not require.

- In the setting with environment variations the formerly learned skills are not updated to work on new environments later in training. While this is not a problem in an *expanding* environment (since all skills will still be useful) it becomes an issue in environments that change in different ways, eg by disabling different legs of the ant agent. Now for a downstream task I would need to know which of the skills are still executable in this unseen downstream environment (eg which leg of the ant is currently disabled) as all other skills will not function properly. Priors works will continuously try to adapt all skills to the current environment and thus do not need this additional skill filtering supervision.


### Execution Weaknesses
The paper's execution has a few weak points, in particular in terms of the experimental evaluation:

- The main quantitative evaluation metric used is some distance between skill's end states in a projected state space. I would argue that this does not necessarily captures the diversity of skill learned to reach these states and a more objective metric is the performance on downstream tasks (since this is what we ultimately care about). While the paper does evaluate downstream learning performance, it is not evaluated on all tasks and in general the experiment seems rushed: it is only presented in the appendix and seemingly only evaluated on a single seed, which is bad practice for RL experiments. I would suggest to instead evaluate all experiments with RL on (multiple) downstream tasks and report those result as the main quantitative evaluation.

- It is unclear to me why the baselines perform that badly on the block environments in Figure 2, particularly the Off-DADS approach. I would expect it to learn still somewhat diverse trajectories in the 2D plane it is optimized to diversify (like it does in Fig 4), but instead it learns skills that always walk in the same direction. The text does not explain this phenomenon clearly. Maybe I am not fully understanding the Off-DADS objective?

- The paper argues that the baselines work better than the proposed method in the cheetah environment since they share parameters, which improves learning efficiency. If this is the case (A) shouldn't we be able to achieve the same performance by training DISk longer? (B) shouldn't parameter sharing also be beneficial for learning efficiency in the harder tasks?

- The method section of the paper does not clearly explain *when* new skills are added. The conclusion alludes to some "convergence criterion", but it would be good to add this to the main section (please correct me if I missed it in the text).


### Further Comments
- The paper follows prior works and uses a hand-defined projection of the state space into a lower-dimensional space in which it measures diversity (eg here 2D velocities for the ant agent). To me this seems like a significant assumption that can not easily be translated to more complicated tasks (what would a simple space for measuring "interestingness" of driving skills look like?). Since this is an assumption that many papers in this line of work make, I won't count it as a weakness for this particular submission.

- The paper's contribution of learning skills sequentially instead of jointly seems more largely applicable than just in the context of continual learning (although continual learning is an obvious and intuitive application). If there is a good explanation for why the incremental learning works better than the joint learning even on static environments, it might strengthen the paper's scope and clarity to present it as a full (improved) alternative to DIAYN / DADS and show the continual learning results only as one additional application that was impossible to solve with prior methods.


## Questions

- Do the baseline approaches that learn skills jointly with weight sharing get the same overall (1) number of environment interactions and (2) number of trainable network parameters?

- I wonder whether the "greedy" approach to skill learning in the proposed method (ie learning one skill after the other as opposed to all jointly) can lead to optimization issues where the first discovered skill is somehow suboptimal (eg by combines two primitives into one skills that are frequently used independently of each other in different parts of the skill space and should thus be learned as separate skills) and this suboptimal skill leads to all subsequently found skills also being suboptimal. To put it another way, I am wondering whether the problem of discovering the most optimal skill set can be solved greedily or whether it requires to be solved as a whole in order to optimally divide the behavior space into skills. I am uncertain about this question, but would be interested to hear the authors thoughts on it.

- The idea of learning separate neural networks in a sequential fashion seems applicable to continual learning problems outside RL -- are there other works that apply this idea beyond RL? It might be useful to add more discussion of approaches that train separate neural networks for continual learning in the related work?

- The paper proposes to use a non-parametric estimate for the state entropy. How does this compare to priors works who also needed to measure entropy? Could we instantiate the proposed method with the techniques used in prior works or in turn instantiate prior works with the used non-parametric estimate to understand the relative contribution of this design choice?

**Summary Of The Paper:**

The paper proposes an approach for unsupervised learning of diverse skills by maximizing an intrinsic reward function. In contrast to prior works (eg DIAYN, DADS) the proposed approach learns skills incrementally using a separate neural network for each skills. In experimental evaluations on OpenAI gym environments this leads to (a) improved diversity in discovered skills and (b) the ability to learn skills in continually changing environments, where prior approaches struggle.


**Summary Of The Review:**

In summary, I think the paper proposes an interesting problem along with a simple solution that is clearly explained. As listed above I do however have some major concerns, both conceptually and in terms of the experimental evaluation, and thus cannot recommend acceptance of the submission in its current form.

Post Rebuttal
============
After considering the authors' rebuttal I have increased my score to recommend (weak) acceptance.

---

> ### Author Response · Authors · 2021-11-12
> **Response To Official Review of Paper1845 by Reviewer N9w1 (3/3)**
>
> ## Response to questions
> * All baselines get the same number of environment interactions over all skills, and the number of network parameters are taken from their official implementations, which In practice meant a wider network for the baselines. Furthermore, our DISk implementation takes 2.5 hours for 1M environment interactions, as opposed to 3.3 for DIAYN and 4.2 for Off-DADS, because we don’t have to train any additional networks (discriminator for DIAYN, forward dynamics model for Off-DADS.)
> * It’s not hard to imagine cases where the “greedy” approach may fail to find the “theoretically optimal” set of skills. However, the latent space coverage that is covered by each skill could still be tuned in practice by tuning the relative weight hyperparameter between the diversity and the consistency objectives.
> * Yes, adding models, both neural and non-neural, in a sequential fashion has indeed been used before in [2-5, 7], with more examples in the book Lifelong Machine Learning [10]. We have updated our Background (Sec. 2) section with such examples.
> * We chose a non-parametric method simply because of the difficulty of dealing with an increasing number of skills in a parametric manner. Even if we were to try to modify prior works with a non-parametric mutual information estimation, they would still learn skills concurrently and thus be vulnerable to the adaptability and the forgetting issues of a changing environment.
>
> Finally, in light of the above clarifications, assuming we have addressed your concerns, we would like to ask if you are willing to increase your score. Otherwise, please let us know if you have additional questions and we would be more than happy to discuss further.
>
> [1] Li, Z., & Hoiem, D. (2017). Learning without forgetting. IEEE transactions on pattern analysis and machine intelligence, 40(12), 2935-2947.
> [2] Ruvolo, P., & Eaton, E. (2013). ELLA: An Efficient Lifelong Learning Algorithm. ICML.
> [3] Ammar, H. B., Eaton, E., Ruvolo, P., & Taylor, M. E. (2015, February). Unsupervised cross-domain transfer in policy gradient reinforcement learning via manifold alignment. In Twenty-Ninth AAAI Conference on Artificial Intelligence.
> [4] Tanaka, F., & Yamamura, M. (1997, August). An approach to lifelong reinforcement learning through multiple environments. In 6th European Workshop on Learning Robots (pp. 93-99).
> [5] Wilson, A., Fern, A., Ray, S., & Tadepalli, P. (2007, June). Multi-task reinforcement learning: a hierarchical bayesian approach. In Proceedings of the 24th international conference on Machine learning (pp. 1015-1022).
> [6] Bendale, A., & Boult, T. E. (2016). Towards open set deep networks. In Proceedings of the IEEE conference on computer vision and pattern recognition (pp. 1563-1572).
> [7] Fei, G., Wang, S., & Liu, B. (2016, August). Learning cumulatively to become more knowledgeable. In Proceedings of the 22nd ACM SIGKDD International Conference on Knowledge Discovery and Data Mining (pp. 1565-1574).
> [8] Fernando, Chrisantha, et al. "Pathnet: Evolution channels gradient descent in super neural networks." arXiv preprint arXiv:1701.08734 (2017).
> [9] Abraham, W. C., & Robins, A. (2005). Memory retention and weight plasticity in ANN simulations. Trends in Neurosciences, 2(28), 73-78.
> [10] Chen, Z., & Liu, B. (2018). Lifelong machine learning. Synthesis Lectures on Artificial Intelligence and Machine Learning, 12(3), 1-207.
> [11] Kalashnikov, D., Varley, J., Chebotar, Y., Swanson, B., Jonschkowski, R., Finn, C., ... & Hausman, K. (2021). MT-Opt: Continuous Multi-Task Robotic Reinforcement Learning at Scale. arXiv preprint arXiv:2104.08212.

---

> > ### Comment · Reviewer_N9w1 · 2021-11-27
> > **Rebuttal Reply**
> >
> > Thank you for your very thorough rebuttal. I appreciate the clarifications and that the hierarchical RL evaluations moved to the main text now. I still think the HRL experiments could be strengthened a bit more: from the current description it is unclear to me what environment setup the evaluation was performed in (eg are all these HRL results run in the no-obstacle Ant environment?) -- the text only mentions what environments were used to train the skills. It would also be good to add a "from scratch" baseline to show that the learned skills actually help over not learning skills at all and to get a better understanding whether they impair final convergence performance in contrast to a flat "from scratch" baseline trained to convergence.
> >
> > That being said, I think the clarifications of the rebuttal and the addition of the HRL experiments to the main paper do improve the submission and thus I am willing to increase my score to vote for acceptance.
> >
> > On a final note, I do not fully agree with the authors' argument about "skill filtering": it is true that all skill-based methods perform HRL over skills to learn which skills to apply when in order to solve the task at hand. Yet, usually we assume that all skills "work" and the question is just which ones are "appropriate" to solve the task. In the discussed setup here we face a changing environment and since the proposed method does not train all skills to work in all environments but instead creates distinct new skills for new environments, there will necessarily be a large set of skills that have never been trained on the target environment and are not even supposed to work there (eg in the case of the ant with different disabled legs). This can pose a greater challenge to downstream RL since it now also needs to figure out which skills even work in the target environment. Alternative continual skill learning approaches might choose to train all skills to work in all environments (thus learning a smaller set of more general purpose skills) so the skill-filtering challenge for downstream RL looks more like in the classic "static environment" setting. Therefore, I wouldn't say that we can just say the challenge is the same for HRL, no matter the design decisions of the continual skill learning method. I hope that this makes the point I was trying to make a little clearer. -- Also note that the current HRL experiments don't investigate this setting since the tested environments fall under the "expanding environment" case I mentioned in my review where all skills still work in the final environment.

---

> > > ### Author Response · Authors · 2021-11-29
> > > **Following up on HRL concerns**
> > >
> > > We appreciate your vote for acceptance, and would like to address your remaining concerns below.
> > >
> > > We are updating our HRL experiment section with more information in the camera ready version and we appreciate your pointers. We ran the HRL experiments in the no-obstacle Ant environments, as you deduced correctly. It takes 5e5 steps for our baseline pure SAC agent to learn to run in a constant direction (so not goal-conditioned) with a dense reward, which is the limit of our plots in the HRL experiment, and we will add this baseline to the HRL plots in the final version.
> > >
> > > Finally, you make an interesting point in:
> > > > Alternative continual skill learning approaches might choose to train all skills to work in all environments (thus learning a smaller set of more general purpose skills) so the skill-filtering challenge for downstream RL looks more like in the classic "static environment" setting.
> > >
> > > We agree with your assessment that an alternative “ideal” algorithm may produce skills in a changing environment where each skill works in all possible variants of the world, like all different sets of broken legs. Compared to such an “ideal” algorithm, DISk falls short by producing new skills that each work with its own set of broken legs.
> > > However, such an ideal algorithm, to the best of our knowledge, does not exist. The baselines we examined in the paper struggle to retain knowledge of past environments instead of fine-tuning the skills to work with all environments.
> > >
> > > Additionally,
> > > > … there will necessarily be a large set of skills that have never been trained on the target environment and are not even supposed to work there (eg in the case of the ant with different disabled legs). This can pose a greater challenge to downstream RL …
> > >
> > > In an environment where invalid actions are just no-ops (for example, a broken-legged ant where a command sent to the broken leg is a no-op, which is our experimental setting,) we maintain that this is a similar challenge as the traditional HRL setting albeit with a slightly larger set of discrete skills. While we agree that this larger possible set of skills, as well as combining skills and changed environments unseen in training, is a marginally larger challenge for HRL, we believe the ability to perform well in changing environments is worthwhile – especially since the prior baselines fail to learn or adapt well at all in such a setting.

---

> ### Author Response · Authors · 2021-11-12
> **Response to Official Review of Paper1845 by Reviewer N9w1 (2/3)**
>
> ## Response to the execution comments:
>
> **Quantitative metric:** Our aim in introducing the mean Hausdorff distance metric is to put commonly used qualitative measures of drawing out trajectories with a quantitative measure that appropriately captures the desirable properties of consistency and diversity. It is not meant as a replacement for the HRL metric, which we have also presented in this work.
>
> **Hierarchical RL metrics** We moved the hierarchical RL experiments to the appendix to maintain the 8-page limit and showcase more of the changing environment experiments. Since we have an extra page for the rebuttal and camera ready, we have added this result back in the main paper. In the static Ant hierarchical environment hierarchical experiments we report our results over three seeds with the standard deviation across them being depicted in shadowed lines. We are currently running experiments to add more seeds throughout all hierarchical experiments.
>
> **Off-DADS in Block environment**: In the Block environment, we believe Off-DADS reaches a local optima of its goal function quickly, and does not escape that optima even after the environment changes further because of lack of exploration. Walking in a single direction appears in the plot because that is the only way out of the circle of blocks at the beginning of training, and Off-DADS optimizes over this direction by walking at velocities different in magnitude. This highlights the need of an algorithm that can continuously evolve as the environment changes and DISk provides an effective solution to this problem.
>
> **Cheetah environment:** As for the Cheetah environment: while parameter sharing could make learning faster for simple environments, our aim is to create a method that can learn robustly in changing environments, which parameter sharing can make difficult. We did not allot more time for training DISk compared to baselines to keep our comparisons fair. Your intuition is correct that given more time DISk does perform better. However, with small amounts of training steps prior work is competitive on this task. DISk is hence most useful for more challenging tasks such as the Ant tasks.
>
> Finally, you asked:
> > The method section of the paper does not clearly explain when new skills are added. The conclusion alludes to some "convergence criterion", but it would be good to add this to the main section (please correct me if I missed it in the text).
>
> In two of our experiments we see effects of different addition criteria. In the broken leg experiment, we add a new skill whenever the environment changes, while in the Blocks experiment we use the same schedule as the static environment and can observe how DISk is affected by different rates of environment change.

---

> ### Author Response · Authors · 2021-11-12
> **Response to Official Review of Paper1845 by Reviewer N9w1 (1/3)**
>
> First of all, we would like to thank you for taking the time to write such a detailed and thoughtful review. We are also glad that you find our work interesting, clear, and simple. We address the questions you have raised in detail below. Since it is a little long, we are splitting it into parts.
>
> ## Response to the conceptual comments:
>
> **Definition of ‘changing world’** We agree that having a standard definition of “changing world” is important and we have added clarification in the paper. We had initially excluded this since in the field of continual learning, a large array of very different approaches has been proposed, but we have updated our Background (Sec. 2) and Related Work (Sec. 5) according to your suggestion to highlight some relevant works. For example, in the field,
> * Some work [1, 6, 7] focuses on learning continually while preserving the original capabilities of the model against catastrophic forgetting,
> * Other works [4, 8] consider a sequence of disjoint tasks, and use earlier tasks to find biases for later tasks, and
> * Finally [2, 3] considers lifelong learning to be a sequence of independent challenges and focus on bidirectional flow of information, where earlier models help the algorithm learn faster in the new tasks, and new tasks help refine old models.
> Furthermore, some works consider environments to be sampled from a distribution [4, 5] while others consider an open-world learning approach [2, 3, 6, 7] where the new tasks may come from anywhere, in any order or number.
> We chose to focus on the *open-world learning approach*, and as a natural result our method is naturally robust to catastrophic forgetting. To demonstrate the generality of DISk, in the Blocks environment, we consider a setting closer to [4, 8], while in the broken leg experiment we consider a setting closer to [1, 6, 7].
>
> **Parameter sharing** In our method, we share the environment interaction data from old skills by relabelling them for the new skills. The effectiveness of pre-training and fine-tuning a policy may indeed improve efficiency, but it is not a common practice in RL and making it work effectively is an area of active research [11]. Thus, we chose to present the simplest effective version of our algorithm. Note that although our method is not sharing parameters, it is sharing data by relabelling past skill experience in the replay buffer. In the end, any method that we choose will be subject to the stability-plasticity dilemma [9] in the lifelong learning setting, and we believe it is more appropriate for future work to hopefully find a better point in the trade-off curve.
>
> **Iterative nature** We believe there must have been a misunderstanding of our experimental setup. In the Blocks experiment, we operate in the setting where DISk has no a priori knowledge of the environment changing. Furthermore we have ablated in Fig. 2 (middle) and shown how faster changes in the environment leads to degradation in performance. Note that prior works perform considerably worse in all settings of this experiment.  In static environments, we simply start training a new skill after the convergence of an old one, and we provide concrete approximate numbers in the appendix for ease of replication. In changing environments, where the rate of change is variable, we have shown that our algorithm is robust even with a fixed skill learning schedule in the Blocks experiment in Section 4.2. The only fundamental limitation seems to be where the environment changes drastically before any skill has a chance to converge; but even humans are susceptible to such dramatic changes.
>
> **Skill filtering** Almost all prior work in skill learning is aimed at a hierarchical setting. Here the final step is fundamentally necessary to be a skill filtering supervision. Almost tautologically, the goal of the task for the hierarchical controller is to provide this skill filtering supervision. Even outside of a hierarchical setting, in DIAYN there is no semantic meaning to the skill index, while Off-DADS has a nominally infinite number of skills. Thus we simply believe that in all skill learning methods mentioned in our paper, we need to know which skills are executable in the downstream environment to be successful. This is precisely how DISk is used in the HRL experiments described in Section 4.4.

---

### Author Response · Authors · 2021-11-12
**Global Comments by Authors of One After Another: Learning Incremental Skills for a Changing World**

We thank the reviewers for their thoughtful and constructive feedback. We have tried to address all of your questions and clarify our paper in individual replies to your review. We have also run several new experiments as suggested by the reviewers to better understand the contributions of this work. However there are some shared serious concerns raised by all reviewers that we believe is a misunderstanding and has been updated in the main paper: *sharing parameters between skills*. We will address this concern in this global comment section due to the shared concern.

**Independent Skill Learning**: Three out of four reviewers have asked for an ablation that uses our baseline such as Off-DADS but with independent skill policies. This is a highly non-trivial task. In fact we believe that one of the contributions of our work DISk is that it can seamlessly use independent skills. So, why is it non-trivial to combine independent skills with off-DADS? Concretely, Off-DADS, is conditioned by continuous skills, which can’t properly be translated without having an infinite class of independent policies. In the original DADS paper Fig. 5 shows the ablation of changing it to discrete independent skills severely reduces the agent performance. The only baseline we use that is compatible with independent skills is DIAYN, and we have since run an ablation study on DIAYN that uses independent skills. The result of this study shows some improvement over DIAYN from the version by Eysenbach et. al. but underperforms Off-DADS or DISk. Nonetheless, we have added this experiment to Appendix G3 of our paper to quell any doubt that integrating independent skills is in itself the primary source of performance improvement.

**Sharing weights in an incremental setting**: One of the major difficulties that led to us coming up with the current incarnation of DISk is not being able to add a new skill in a shared-parameter skill-conditioned policy network like in the prior methods. We experimented with transferring weights from old skills to new ones, adding new heads to a wide, shared network, but the architecture that worked the best was just initializing a new network, which we show in the paper. This is intuitively in line with results in several recent papers that show that sharing weights in RL requires careful consideration and is in itself an active area of research [1].

To reiterate, DISk proposes two key ideas: incremental learning and independent skills. By itself, neither is sufficient. Incremental learning on shared skills leads to catastrophic forgetting as highlighted in the ablation study in Section 4.2. Independent skillset without incremental learning is infeasible to effectively integrate with prior state of the art. What makes DISk special is that the combination of these two ideas fixes the gap that would otherwise exist in their isolated application. Furthermore to the best of our knowledge, DISk is the **only skill-based approach that can tackle non-stationary environments**.

[1] Kalashnikov, D., Varley, J., Chebotar, Y., Swanson, B., Jonschkowski, R., Finn, C., ... & Hausman, K. (2021). MT-Opt: Continuous Multi-Task Robotic Reinforcement Learning at Scale. arXiv preprint arXiv:2104.08212.

---

### Author Response · Authors · 2021-11-16
**Revision Summary**

We thank the reviewers for their detailed comments. We have responded to the valuable comments by our reviewers by making a number of modifications to address their concerns and suggestions for improvements. A summary of the changes we made in the paper is shared below:

1. As suggested by Reviewer bA27, We have added references for our statement “.. the agents generalize poorly to any changes in the environment” in the introduction.
2. Addressing the comments by Reviewer N9w1 and eZE6, we have added a section about incremental/lifelong learning in our “Backgrounds and Preliminaries” (Sec. 2), explaining three different standard problem setups, and standard practices like growing model parameters over time.
3. Per Reviewer eZE6’s suggestion, we updated our mathematical notation on Sec. 3.2 “A Practical Algorithm” to draw further distinction between random variables and the distributions they are drawn from, and make our notation more mathematically sound.
4. To address a confusion pointed out by Reviewer eZE6, we have added inspirations for our models for continuous environment changes in Sec. 4.2 “Can DISk Adaptively Discover…”.
5. Answering a question by Reviewer bA27, we have added a new figure in Appendix Fig. 8, and referenced it from Sec. 4.2, which further elucidates that with DADS in a Broken leg environment, catastrophic forgetting is a bigger challenge than lack of adaptation.
6. Acknowledging Reviewer N9w1’s point, we have moved the Hierarchical RL experiment figure from Appendix to the main paper as Fig. 5, making the HRL experiments a larger part of our paper.
7. Responding to suggestions by Reviewers N9w1, eZE6, and HdqY, we have added further discussion pertaining to weight sharing between skills, and its relation to the nature of DISk and the baselines in the “Ablation Analysis” (Sec. 4.5), specifically how it is nontrivial to have a shared-skill version of DISk.
8. Following up on the above point for the same Reviewers N9w1, eZE6, and HdqY, we have added a baseline ablation experiment in the Appendix (App. G.3) with a disjoint skill variant of a baseline (DIAYN).
9. Addressing some comments by Reviewer N9w1 and eZE6, we have added more references, mostly on canonical works in lifelong RL, to the Incremental Learning paragraph in “Related Works”, Sec. 5.
10. Thanks to Reviewer eZE6, we have added references to the unsupervised learning part of the related work section, namely VALOR by Florensa et. al. (2017) and Diverse Successor Policies by Zahavy et. al. (2021).
11. We have clarified our training schedule in App. D4 in response to questions by Reviewer N9w1 and eZE6.

---

> ### Author Response · Authors · 2021-11-22
> **Additional revision**
>
> 12. At Reviewer eZE6's suggestion, we have added a new ablation experiment where DIAYN is trained with a VALOR-like (Achiam et. al. 2018) expanding skill schedule and presented the result in Appendix G4.

---

### Decision · Program_Chairs · 2022-01-20

**Decision:**

Accept (Poster)

**Comment:**

This paper is about an unsupervised method to learn new skills in non-stationary environments by maximizing an intrinsic reward function. Experimental evaluations on OpenAI gym environments show that the proposed approach improves the diversity of the learned skills and is able to adapt to continuously changing environments.

This paper is borderline. After reading each other's reviews and the authors' feedback, the reviewers discussed the pros and cons of this work. Even if the reviewers have pointed out that the paper has some limitations, they agree that the paper represents a valuable contribution and have appreciated the improvements implemented by the authors during the rebuttal, thus reaching a consensus towards acceptance.
The authors need to update their paper according to what they have proposed in their response and they have to take into serious considerations all the reviewers' suggestions while they will prepare the camera-ready version of their paper.